# ROUTE EXPERTS BY SEQUENCE, NOT BY TOKEN

## ABSTRACT

Mixture-of-Experts (MoE) architectures scale large language models (LLMs) by activating only a subset of experts per token, but the standard *TopK* routing assigns the same fixed number of experts to all tokens, ignoring their varying complexity. Prior adaptive routing methods introduce additional modules and hyperparameters, often requiring costly retraining from scratch. We propose **Sequence-level TopK (SeqTopK)**, a minimal modification that shifts the expert budget from the token level to the sequence level. By selecting the top $T \cdot K$ experts across all $T$ tokens, SeqTopK enables end-to-end learned dynamic allocation – assigning more experts to difficult tokens and fewer to easy ones – while preserving the same overall budget. SeqTopK requires only a few lines of code, adds less than 1% overhead, and remains fully compatible with pretrained MoE models. Experiments across math, coding, law, and writing show consistent improvements over TopK and prior parameter-free adaptive methods, with gains that become substantially larger under higher sparsity (up to 16.9%). These results highlight SeqTopK as a simple, efficient, and scalable routing strategy, particularly well-suited for the extreme sparsity regimes of next-generation LLMs.

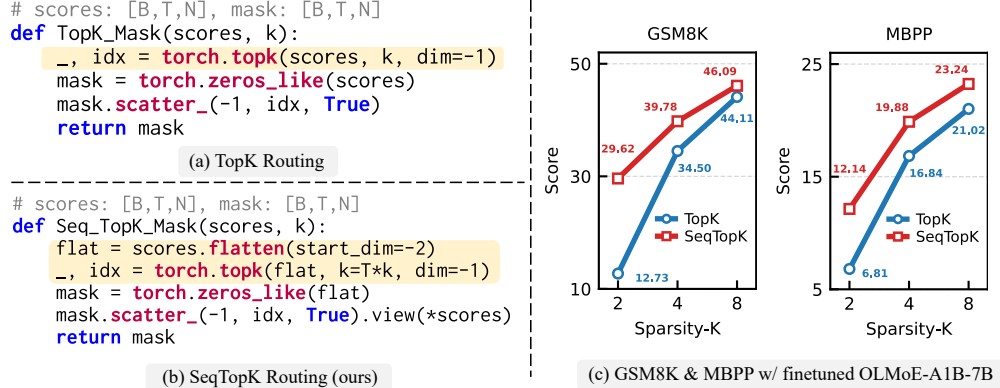

```
# scores: [B,T,N], mask: [B,T,N]
def TopK_Mask(scores, k):
    _, idx = torch.topk(scores, k, dim=-1)
    mask = torch.zeros_like(scores)
    mask.scatter_(-1, idx, True)
    return mask
```

(a) TopK Routing

```
# scores: [B,T,N], mask: [B,T,N]
def Seq_TopK_Mask(scores, k):
    flat = scores.flatten(start_dim=-2)
    _, idx = torch.topk(flat, k=T*k, dim=-1)
    mask = torch.zeros_like(flat)
    mask.scatter_(-1, idx, True).view(*scores)
    return mask
```

(b) SeqTopK Routing (ours)

(c) GSM8K & MBPP w/ finetuned OLMoE-A1B-7B

Figure 1: **Overview of our proposed method.** **(a) & (b)** Simple PyTorch implementations of TopK and SeqTopK routing. With a minimal modification of TopK routing (highlighted), SeqTopK routes experts via comparing expert scores across all tokens in a sequence, enabling dynamic and context-aware allocation of experts (e.g., more experts for hard tokens) via end-to-end training. **(c)** Performance of fintuned OLMoE-A1B-7B on GSM8K and MBPP datasets. SeqTopK consistently outperforms TopK under different expert budgets ($K = 2, 4, 8$), and the gain is much larger under sparser MoEs.

## 1 INTRODUCTION

Mixture-of-Experts (MoE) architectures have emerged as a central paradigm for scaling large language models (LLMs), offering massive capacity without a proportional increase in computation (Shazeer et al., 2017; Lepikhin et al., 2020; Fedus et al., 2022). By activating only a small subset of experts per token, MoEs distribute knowledge across specialized modules while maintaining efficiency. However, the dominant routing strategy – *TopK selection* – treats all tokens

uniformly, assigning each token the same fixed number of experts ($K$) regardless of its complexity or information content. This rigidity overlooks the contextual heterogeneity of language (Smith & Levy, 2013): depending on the semantic context, some tokens are trivial and require little capacity (e.g.,"the"), while others (e.g., legal terms) are far more informative and benefit from broader expert consultation.

As a result, computation is wasted on easy tokens, while complex tokens receive insufficient expert allocation under TopK routing.

To address this limitation, a growing body of work has explored adaptive routing strategies (Huang et al., 2024; Lu et al., 2024; Wang et al., 2024c; Guo et al., 2024; Wu et al., 2025; Jin et al., 2024a). While promising, these approaches often introduce additional modules and hyperparameters, and typically require pretraining the MoE model from scratch with the new routing. As a result, they come with substantial cost, architectural complexity, and limited large-scale validation. This raises a natural question: *do we truly need entirely new routing mechanisms to enable adaptive allocation?*

In this paper, we instead revisit standard TopK routing and propose a simple but effective relaxation: shifting the budget from the *token level* to the *context level*. Instead of enforcing $K$ experts per token, we allocate the top $T \cdot K$ experts across all $T$ tokens in a sequence. This *Sequence-level TopK* (SeqTopK) routing enables the model, through end-to-end likelihood maximization, to assign more experts to challenging tokens and fewer to trivial ones, thereby achieving dynamic allocation without introducing new architectural components. Crucially, SeqTopK preserves the same overall budget as TopK, requires only a few lines of code to implement (Figure 1), and introduces no additional parameters or hyperparameters. It is fully compatible with existing MoE models and can be directly fine-tuned from pretrained TopK checkpoints. SeqTopK also integrates naturally with autoregressive decoding via an on-the-fly updated list of expert scores, which we term the *Expert Cache* (analogous to the KV cache in attention). Overall, SeqTopK incurs less than 1% overhead in computation and memory, while seamlessly extending the capabilities of existing MoE implementations.

Despite its simplicity, SeqTopK consistently outperforms standard TopK and prior adaptive methods. The gains are especially pronounced under higher sparsity (up to 16.89%), highlighting its promise as state-of-the-art MoE models, such as GPT-OSS (Agarwal et al., 2025) and Qwen-Next (Yang et al., 2025), continue to scale with extreme sparsity. We validate SeqTopK across diverse domains, including math, coding, law, and writing, and demonstrate its data-driven adaptive behavior that allocates more experts to harder tokens.

Our contributions are summarized as follows:

- We propose **SeqTopK**, a routing mechanism that maintains the same compute budget as the TopK routing while enabling context-level dynamic expert allocation.
- For efficient autoregressive inference, we introduce **online SeqTopK with Expert Cache**, which maintains an updated list of expert scores to support online routing that closely mirrors training-time SeqTopK.
- We provide extensive validation across multiple downstream tasks, showing that SeqTopK consistently outperforms the standard TopK and delivers larger gains as sparsity increases, along with qualitative evidence of its context-aware dynamic routing behavior.

## 2 BACKGROUND

**Mixture of Experts (MoEs).** We first introduce a generic MoE architecture commonly used in Transformer-based language models (LMs) (Vaswani et al., 2017). A standard decoder-only transformer (Radford et al., 2019) comprises $L$ layers, where each block can be represented as follows:

$$\boldsymbol{x}_t^l = \text{Self-Attn}(\boldsymbol{h}_{1:t}^{l-1}) + \boldsymbol{h}_{1:t}^{l-1}, \tag{1}$$

$$\boldsymbol{h}_t^l = \text{FFN}(\boldsymbol{x}_t^l) + \boldsymbol{x}_t^l, \quad t \in [T], \tag{2}$$

where $T$ denotes the length of the sequence $x_{1:T}$, Self-Attn($\cdot$) and FFN($\cdot$) are self-attention and the Feed-Forward Network. Here, $\boldsymbol{x}_t^l \in \mathbb{R}^D$ is the hidden state of the $t$-th token and $\boldsymbol{h}_t^l \in \mathbb{R}^D$ the output of the $l$-th transformer block. From now on, we drop the layer index $l$ for better clarity.

A typical approach for constructing MoE language models involves substituting FFNs in transformers with MoE layers. These MoE layers partition the FFN into $N$ smaller FFNs, $E_1, \ldots, E_N$,

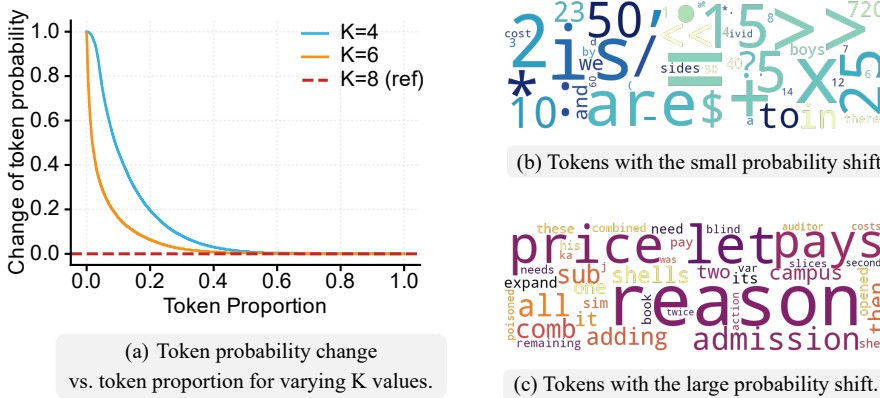

(b) Tokens with the small probability shift.

(a) Token probability change vs. token proportion for varying K values.

(c) Tokens with the large probability shift.

Figure 2: **Change of token probability $P(\boldsymbol{x}_t|\boldsymbol{x}_{<t})$ under varying active experts ($K$). (a) Token probability change vs. token proportion.** Given the same prefix, we sample over 10k tokens across different K values, computing probability differences relative to $K = 8$ (the original OLMoE setting). About 60% of tokens show little probability change when $K$ is reduced from 8 to 4 or 6, while 10% change dramatically, indicating that different tokens can require quite different numbers of activated experts to predict. **(b) & (c) Word clouds of the top 50 tokens with small ($< 0.01$) and large ($> 0.5$) token probability shifts.** Tokens with the larger probability differences are content words that influence semantic direction or topic shifts, whereas tokens with the smaller probability differences are numbers or function words that maintain structure.

referred to as *experts*, where $N$ denotes the total number of experts. Subsequently, each token $x_t$ is assigned to a few experts as determined by a router module $R(\cdot)$. The output is computed as:

$$\boldsymbol{h}_t = \sum_{i=1}^{N} R_i(\boldsymbol{x}_t) E_i(\boldsymbol{x}_t) + \boldsymbol{x}_t. \tag{3}$$

**Standard TopK Routing in MoEs.** TopK routing (Shazeer et al., 2017; Lepikhin et al., 2020; Fedus et al., 2022) is a widely used, de facto strategy for routing experts in modern LLMs. For each input token $x_t$, the TopK router identifies the $K$ most relevant experts and sets the relevance scores of all other experts to zero, i.e.,

$$(\text{scores}) \quad \boldsymbol{s}_t = \sigma(\boldsymbol{W}\boldsymbol{x}_t), \boldsymbol{W} \in \mathbb{R}^{N \times D}; \tag{4}$$

$$(\text{TopK routing}) \quad R_i(\boldsymbol{x}_t) = \begin{cases} 1, & \text{if } i \in \arg\text{TopK}_{i \in [N]} \ \boldsymbol{s}_{t,i} \\ 0, & \text{otherwise}, \end{cases} \tag{5}$$

where $\sigma(\cdot)$ denotes the *softmax* function that normalizes the scores over the $N$ experts, and $\arg\text{TopK}(\cdot)$ returns the index set of the $k$ largest elements along the chosen dimension[1].

Due to the limited space, we defer a detailed discussion on other related works to Appendix A.

## 3 METHOD

### 3.1 SEQTOPK: FROM TOKEN-LEVEL TO SEQUENCE-LEVEL ROUTING

**Token Heterogeneity in Standard TopK.** While standard TopK routing adaptively selects experts for each token, it enforces a *fixed budget of $K$ experts per token*, regardless of predictive difficulty. This assumption ignores the heterogeneity among tokens in a sequence. As shown in Figure 2, when reducing the routing budget from $K = 8$ to $K = 4$, over 60% of tokens exhibit *little or no degradation* in their predicted likelihood $P(\boldsymbol{x}_t \mid \boldsymbol{x}_{<t})$, whereas approximately 15% of tokens suffer

---

[1]Since the primary distinction between TopK and SeqTopK lies in expert allocation, we omit the final routing-weight selection step in Eq. 5 (*i.e.*, selecting weights from $s_t$ based on the chosen indices) for simplicity.

a *drop of more than* $0.5$. Theoretically, we prove in Section E that such uniform budget allocation is inherently suboptimal. Collectively,this evidence highlights substantial variation in token difficulty and the computational resources required. This suggests that standard TopK over-serves easy tokens and under-serves difficult ones, leading to inefficiency and degraded accuracy in sparse regimes.

**Limitation of Token-level Routing.** Several recent works attempt adaptive routing by replacing TopK with learnable thresholding functions (e.g., ReLU gating in ReMoE or identity experts (Jin et al., 2024a; Team et al., 2025b)), thereby allowing different tokens to activate different numbers of experts. However, such methods show limited performance gains because they remain fundamentally at the local *token level*: routing decisions depend only on each token's own scores, without considering the surrounding context. In contrast, the power of self-attention in Transformers arises precisely from *contextualization*: predictions are informed by comparing a token with its neighbors. MoE layers that treat tokens independently lack this context-aware adaptivity.

**Sequence-level Routing with SeqTopK.** Motivated by these observations, we propose **SeqTopK**, a context-aware routing strategy that performs expert selection at the sequence level. Let

$$\text{(SeqTopK scores)} \quad \boldsymbol{S} = [\boldsymbol{s}_1, \ldots, \boldsymbol{s}_T]^\top \in \mathbb{R}^{T \times N}, \boldsymbol{s}_t = \sigma(\boldsymbol{W}\boldsymbol{x}_t) \in \mathbb{R}^N, \tag{6}$$

where $\boldsymbol{S}$ collects the expert scores for all tokens in a sequence of length $T$ with $N$ experts. SeqTopK selects the top $K_{\text{seq}}$ entries across all $T \cdot N$ elements in the score matrix $\boldsymbol{S}$:

$$\text{(SeqTopK routing)} \quad R_i(\boldsymbol{x}_t) = \begin{cases} 1, & \text{if } (t, i) \in \arg \text{TopK}_{t \in [T], i \in [N]} \boldsymbol{S}_{t,i}, \\ 0, & \text{otherwise.} \end{cases} \tag{7}$$

Here $K_{\text{seq}} = T \cdot K$, where $K$ is the original budget of TopK and $T$ is the number of tokens in a sequence, which ensures the total budget of SeqTopK exactly matches that of TopK. Thus, with equal overall compute, tokens can now receive different numbers of experts. Empirically, SeqTopK allocates more experts to difficult tokens and fewer to trivial ones, improving performance without increasing cost. Importantly, the allocation depends not only on a token's absolute score but also on its *relative importance within the sequence*, thereby enabling context-aware routing in MoEs.

**Balanced Routing with Token-level Bounds.** To prevent degenerate allocations (e.g., assigning no experts to some tokens), we enforce per-token bounds for the number of experts. Specifically, each token is guaranteed at least one expert, and capped at $K_{\text{tok}} + 2$ experts, where $K_{\text{tok}}$ is the original token-level sparsity. This ensures all tokens contribute to MoE training while avoiding domination by a small subset of tokens. According to ablation studies in Section D, we find that while SeqTopK's performance exhibits slight fluctuations depending on the specific upper bound chosen, the bounded variant consistently yields robust improvements across most of the evaluated settings.

**Compatibility with TopK MoE models.** While many sophisticated designs for context-aware routing could be envisioned, we deliberately designed SeqTopK to remain closely aligned with standard TopK. At the implementation level, it essentially alters only the indexing dimensions of the TopK operator, *requiring just a few lines of code to modify* (Figure 1). This simplicity yields several practical advantages. First, no additional hyperparameters or auxiliary modules are introduced. Second, pretrained TopK MoE checkpoints can be readily adapted to SeqTopK with only a few hundred fine-tuning steps (Section 4.1). Third, the computational and memory overhead introduced is negligible, typically less than $1\%$ in our profiling. Taken together, these properties make SeqTopK a practical, lightweight drop-in replacement for standard TopK routing, enabling MoEs to perform context-aware adaptive routing and deliver improved performance across diverse downstream tasks.

## 3.2 ONLINE SEQTOPK: EFFICIENT DECODING WITH CACHED EXPERT SCORES

While SeqTopK seamlessly integrates into the training stage as a drop-in replacement for TopK, applying it directly at inference is problematic. Large language models generate tokens autoregressively from $t = 1$ to $T$, yet vanilla SeqTopK requires access to the routing scores of all tokens $\boldsymbol{x}_{1:T}$. In other words, vanilla SeqTopK is inherently *non-causal*[2], and thus makes it incompatible with autoregressive decoding where $\boldsymbol{x}_{m+1:T}$ is unavailable at step $m$.

---

[2]Although non-causal, SeqTopK does not actually leak differentiable information about future tokens, since the $\arg \text{TopK}$ operator in Eq. 7 is non-differentiable.

**Online SeqTopK.**  To address this limitation, we propose **Online SeqTopK**, which restricts expert selection at step $m$ to the $m$ tokens generated so far:

$$\text{(online SeqTopK scores)} \quad \boldsymbol{S}_m = [\boldsymbol{s}_1, \ldots, \boldsymbol{s}_m]^\top \in \mathbb{R}^{m \times N}, \quad \boldsymbol{s}_t = \sigma(\boldsymbol{W}\boldsymbol{x}_t) \in \mathbb{R}^N, \quad (8)$$

$$\text{(online SeqTopK routing)} \quad R_i(\boldsymbol{x}_m) = \begin{cases} 1, & \text{if } (t,i) \in \arg\text{TopK}_{t \in [m], i \in [N]} \boldsymbol{S}_{t,i}, \\ 0, & \text{otherwise.} \end{cases} \quad (9)$$

Here we set the budget to $K_{\text{seq}} = m \cdot K$, so that at most $m \cdot K$ experts are activated across the first $m$ tokens. When $m = T$, Online SeqTopK recovers the vanilla formulation in Eq. 7. Unlike training-time SeqTopK, however, the online variant cannot anticipate future tokens: once an expert assignment is made for earlier tokens, it is fixed and unaffected by later routing scores. Nevertheless, Online SeqTopK guarantees that for any $1 \le m \le T$, the cumulative number of activated experts never exceeds $m \cdot K$, and is thus always bounded by the $T \cdot K$ experts of the vanilla formulation. This property establishes a natural upper bound on computation for efficient autoregressive decoding, while in expectation yielding the same total number of activated experts as the offline variant.

**Expert Cache.**  To avoid recomputing expert scores at each step, we introduce an **Expert Cache** that stores all previously computed scores and is updated only with the scores at the latest step:

$$\text{(Expert Cache)} \quad \boldsymbol{S}_m = \begin{pmatrix} \boldsymbol{S}_{m-1} \\ \boldsymbol{s}_m^\top \end{pmatrix}, \quad \boldsymbol{s}_m = \sigma(\boldsymbol{W}\boldsymbol{x}_m) \in \mathbb{R}^N. \quad (10)$$

This design is analogous to the KV cache in self-attention: whereas the KV cache stores intermediate keys and values, the expert cache maintains routing scores for reuse. In practice, the expert cache can be co-located with the KV cache and managed with identical read/write mechanisms. Its memory footprint is also modest: while the KV cache has size $B \times T \times H$ (with hidden dimension $H \approx 4096$), the expert cache is only $B \times T \times N$ (with $N \in [8, 128]$ in typical MoE models). Consequently, Online SeqTopK introduces negligible memory and compute overhead.

### 3.3  Comparison with BatchTopK

A closely related variant of SeqTopK is BatchTopK (Bussmann et al., 2024), which selects experts across all tokens in a batch rather than within a single sequence. While effective in MoE-based diffusion image modeling, where batches are regular and homogeneous (Yuan et al., 2025; Shi et al., 2025), BatchTopK is ill-suited for language modeling. This is because text sequences in a batch vary in length and domain, making cross-sequence competition unfair and unstable. In deployment, batch sizes are flexible and unpredictable, creating mismatches between training and inference when routing depends on global batch composition. Moreover, selecting experts across independent user inputs risks information leakage, raising privacy and safety concerns in multi-tenant systems. In contrast, SeqTopK restricts competition to tokens within the same sequence, ensuring contextual consistency, robustness to batch size, and data privacy, making it better suited for adaptive routing in language modeling. We will show soon that it yields consistent gains over BatchTopK in practice.

## 4  Benchmark Results and Analysis

In this section, we evaluate our proposed SeqTopK routing strategy from both pre-training and fine-tuning perspectives. Section 4.1 presents fine-tuning results for modern language MoE models under varying sparsity levels across tasks, including mathematical reasoning (Cobbe et al., 2021), code generation (Austin et al., 2021; Chen et al., 2021), summarization, and legal assistance (Wang et al., 2024b). Section 4.2 reports GPT-2–level MoE pre-training results and zero-shot performance on benchmark datasets. A detailed efficiency analysis is presented in Section 4.3, examining the additional training and inference costs of SeqTopK.

### 4.1  Scaling Model Sparsity at Fine-Tuning Stage

**Experiment Setup.**  We select MRL-TopK (Bussmann et al., 2025), BatchTopK (Bussmann et al., 2024), and standard TopK as baselines because they **enable MoE model fine-tuning** while **only varying in their routing mechanism** and requiring **no additional parameters.** We fine-tuned two

Table 1: **Benchmark results of models fine-tuned from the OLMoE-A1B-7B model (64 routing experts) with different *routing methods*.** SeqTopK outperforms baseline methods across all settings, with gains becoming much larger (up to 7.55%) as sparsity increases. Best in **bold** .

| *OLMoE-A1B-7B* Routing Methods | K | Sparsity Ratio | GSM8k 0-shot EM | MBPP 3-shot Pass@1 | HumanEval 0-shot Pass@1 | Summary 0-shot Score | Law 0-shot Score | Avg - Score |
|---|---|---|---|---|---|---|---|---|
| Base | 8 | 1/8 | 15.58 | 19.80 | 10.97 | 7.49 | 5.70 | 11.91 |
| TopK | 8 | 1/8 | 44.11 | 21.04 | 13.41 | 45.31 | 24.89 | 29.74 |
| MRL-TopK | | | 43.93 | 21.78 | 12.21 | 44.25 | 21.08 | 28.65 |
| BatchTopK | | | 44.80 | 22.63 | 14.02 | 42.89 | 22.89 | 29.45 |
| **SeqTopK** | | | **46.09** | **23.21** | **15.24** | **46.40** | **26.52** | **31.49** |
| TopK | 4 | 1/16 | 36.50 | 16.84 | 10.98 | 41.42 | 17.77 | 24.70 |
| MRL-TopK | | | 27.44 | 17.82 | 9.76 | 39.89 | 12.21 | 21.42 |
| BatchTopK | | | 35.93 | 18.62 | 12.82 | 39.79 | 17.82 | 24.99 |
| **SeqTopK** | | | **39.78** | **19.88** | **13.41** | **43.00** | **20.20** | **27.25** |
| TopK | 2 | 1/32 | 12.73 | 6.81 | 6.12 | 37.89 | 8.91 | 14.49 |
| MRL-TopK | | | 2.2 | 4.2 | 4.8 | 19.6 | 3.4 | 6.84 |
| BatchTopK | | | 18.49 | 6.60 | 6.71 | 37.18 | 11.89 | 16.17 |
| **SeqTopK** | | | **29.62** | **12.14** | **12.22** | **42.60** | **14.10** | **22.04** |

Table 2: **Benchmark results of models fine-tuned from the Qwen1.5-MoE-A2.7B model with different *routing methods*.** This model by default, uses 4 shared experts and then selects 4 experts out of 60 routing experts. SeqTopK consistently outperforms the others. Best in **bold.**

| *Qwen1.5-MoE-A2.7B* Methods | K | Sparsity- Ratio | GSM8k 0-shot EM | MBPP 3-shot Pass@1 | HumanEval 0-shot Pass@1 | Summary 0-shot Score | Law 0-shot Score | Avg - Score |
|---|---|---|---|---|---|---|---|---|
| Base | 4 | 1/8 | 38.69 | 38.84 | 32.31 | 28.29 | 18.20 | 31.27 |
| TopK | 4 | 1/8 | 55.16 | 35.02 | 36.89 | 39.21 | 42.32 | 41.72 |
| MRL-TopK | | | 54.88 | 35.76 | 36.13 | 38.74 | 40.25 | 41.15 |
| BatchTopK | | | 53.92 | 35.24 | 35.89 | 37.10 | 42.39 | 40.91 |
| **SeqTopK** | | | **55.87** | **36.41** | **37.20** | **41.31** | **45.29** | **43.19** |

models (Muennighoff et al., 2024; Team, 2024) and evaluated them on five downstream datasets using the framework of Feng et al. (2023). We then scaled the models to extreme sparsity (*e.g.*, reducing the number of activated experts per token from 8 to 2) and reported performance across different sparsity levels. Zero-shot evaluations were conducted on GSM8K (Cobbe et al., 2021), HumanEval (Chen et al., 2021), Summary (Wang et al., 2024b), and Law (Wang et al., 2024b), while 3-shot evaluations were performed on MBPP (Austin et al., 2021). Following standard training procedures, we pad samples to 4096 tokens. Sequences exceeding the context window are truncated to retain the initial segment and preserve primary context Note that BatchTopK is sensitive to the evaluation batch size (see Section 4.4). Therefore, we report the best performance in Table 1 and Table 2. Detailed settings are provided in Section B.2.

**Analysis of default sparsity results.** Table 1 and Table 2 compare OLMoE-A1B-7B (Muennighoff et al., 2024) and Qwen1.5-MoE-A2.7B (Team, 2024) across five downstream tasks under different routing strategies. Base denotes the original model without fine-tuning, while *TopK* and *SeqTopK* represent models fine-tuned with standard TopK routing and the proposed SeqTopK routing, respectively. Under default sparsity level (*i.e.*, activate 8 out of 128 experts for OLMoE and 4 out of 60 experts for Qwen1.5), SeqTopK delivers average performance gains of **5.9%** for OLMoE and **3.6%** for Qwen1.5, demonstrating the effectiveness of its intrinsic context-aware routing. Notably, these improvements are obtained with only minimal code changes, yet SeqTopK consistently surpasses standard routing strategies across tasks. The smaller gain for Qwen1.5 can be attributed to its architecture, in which four of the eight experts are shared across tokens, limiting allocation flexibility

Table 3: **Zero-shot performance of 182M-sized MoE models pretrained with different *routing methods***, where ReMoE uses ReLU as an adaptive routing function. SeqTopK shows best performance at each sparsity level; and comparing the two levels, SeqTopK excels $K = 4$ while others excel at $K = 8$, indicating that SeqTopK benefits more from higher MoE sparsity. Best in **bold** .

| Method | Experts | K | Sparsity-Ratio | LAMBDA | RACE | ARC-E | ARC-C | Avg. |
|--------|---------|---|----------------|--------|------|-------|-------|------|
| TopK | | | | 59.30 | 27.33 | 46.66 | 20.73 | 38.51 |
| ReMoE | 128 | 4 | 1/32 | 59.94 | 28.01 | 47.71 | 21.07 | 39.18 |
| **SeqTopK** | | | | **60.82** | **28.70** | **48.07** | **23.41** | **40.25** |
| TopK | | | | 60.15 | 27.61 | 47.19 | 20.06 | 38.75 |
| ReMoE | 64 | 8 | 1/8 | **61.73** | 28.15 | **49.12** | 19.06 | 39.52 |
| **SeqTopK** | | | | 60.67 | **28.55** | 48.42 | **22.36** | **40.00** |

Table 4: **Efficiency Analysis for SeqTopK.** (a) We report the GPU hours for pre-training a 182M-size model over 30B tokens using various routing strategies. (b) We benchmark the OLMoE-A1B-7B model's decoding efficiency for online SeqTopK and the overhead introduced by Expert-Cache, using *huggingface.gen()* for fair comparison.

(a) Pre-training Time.

| Method | Wall-clock Time (GPU hours) |
|--------|------------------------------|
| TopK | 244.2 |
| ReMoE | 245.0 |
| **SeqTopK** | 246.4 **(+1%)** |

(b) Inference throughput and peak memory use.

| Method | Tokens/s | Peak Memory [GB] |
|--------|----------|------------------|
| TopK | 141.23 | 18.21 |
| **SeqTopK** | 139.41 **(-1%)** | 18.35 **(+1%)** |

compared to OLMoE, which has no shared experts. Unlike prior methods that require pre-training from scratch (Jin et al., 2024a; Wang et al., 2024c; Guo et al., 2024), SeqTopK plugs into pre-trained model fine-tuning, delivering low-overhead performance gains for real-world deployment. In Section C, we study applying SeqTopK decoding on top of TopK-fine-tuned models, showing that our approach works well with modern MoE frameworks.

**Analysis of scaling sparsity results.** As shown in Table 1, we compare routing strategies across varying sparsity levels. MRL-TopK exhibits marked degradation as sparsity increases and underperforms even under the default sparsity setting, suggesting inefficiencies in dynamic expert allocation. By contrast, BatchTopK can match the performance of standard TopK across sparsity levels, but it is highly sensitive to the evaluation batch size, which limits its practical deployment (see Section 4.4). Our proposed SeqTopK consistently outperforms baselines under all settings. As sparsity increases (*i.e.*, $K$ decreases from 8 to 2), the performance gap widens. This trend suggests that SeqTopK's context-aware expert allocation is particularly advantageous in high-sparsity regimes, where effective utilization of limited expert capacity is critical. Under the extreme regime ($K = 2$), SeqTopK achieves nearly **twice** the performance of the standard TopK strategy, further underscoring its strong ability to handle ultra-sparse settings via intrinsic dynamic routing. This is a fair comparison, as none of the evaluated models are pre-trained under such extreme sparsity conditions. The substantial performance lead of SeqTopK suggests that it is particularly well-suited for ultra-sparse models in both pre-training (see Section 4.2) and fine-tuning stages.

### 4.2 GPT2-LEVEL PRE-TRAINING EVALUATION

**Experiment Setup.** We select ReMoE (Wang et al., 2024c) (ReLU-routed) and standard TopK serve as our baselines, distinguished **only by their routing mechanism**. Other related work, such as MoE++ (Jin et al., 2024a) and DynMoE (Guo et al., 2024), that incorporate additional zero or gate experts are excluded. For a detailed discussion about these works, see Section A. Following Wang et al. (2024c), we train the models on The Pile (Gao et al., 2020) for 60k steps (~30B tokens), which exceeds the compute-optimal dataset size predicted by Krajewski et al. (2024) and is sufficient for convergence. We follow Megatron-LM (Shoeybi et al., 2019) by concatenating documents via <eod> and slicing them into fixed-length sequences (*e.g.*, 1024). We further employ Generalized

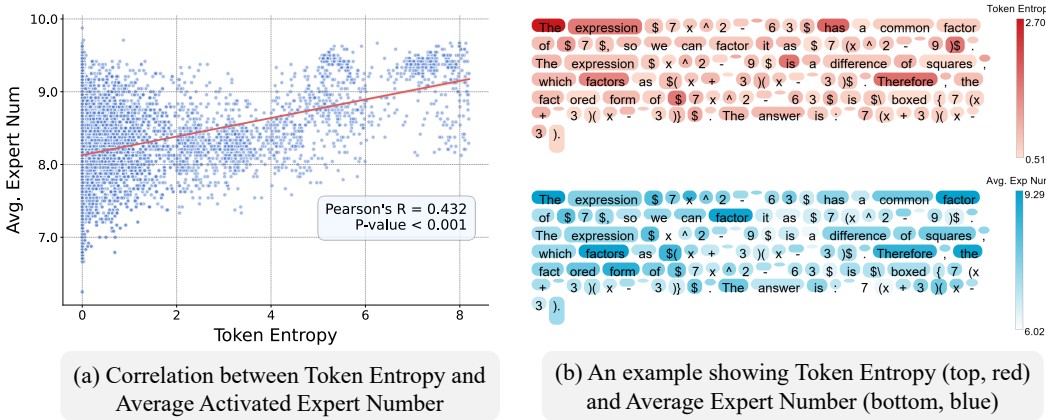

(a) Correlation between Token Entropy and Average Activated Expert Number

(b) An example showing Token Entropy (top, red) and Average Expert Number (bottom, blue)

Figure 3: **Correlation between token entropy and expert activation.** (a) Analysis of 10K tokens generated by fine-tuned Qwen-1.5-A2.7B, showing that higher token entropy—defined as the entropy of the output distribution at each token, capturing prediction uncertainty and token hardness—correlates well with a larger number of average activated experts. (b) Illustration from a specific generated sequence, where SeqTopK often activates more experts on high-entropy tokens, such as the "$" symbol marking the start of a math expression.

Attention Masks and Position ID resetting to guarantee independence between packed documents. We evaluated their zero-shot performance across three downstream tasks: LAMBADA (Paperno et al., 2016); RACE (Lai et al., 2017); ARC (Clark et al., 2018). A more detailed experiment setup is provided in Section B.1.

**Analysis.** Table 3 reports the zero-shot accuracy of different routing methods on downstream tasks across sparsity levels. SeqTopK consistently outperforms the baselines, achieving a notable gain of **3.2%** and **4.5%** over TopK across sparsity regimes with only 1% additional overhead. As sparsity increases, the performance margin widens, indicating that SeqTopK better handles high-sparsity settings through its sequence-level routing. Compared with ReMoE (Wang et al., 2024c), which employs a three-stage training pipeline and is sensitive to sparsity-controlling hyperparameters, SeqTopK integrates seamlessly with existing MoE models, enabling on-the-fly dynamic routing with only a few lines of code.

## 4.3 EFFICIENCY ANALYSIS

We report the GPU hours for pre-training a 182M-size model over 30B tokens using various routing strategies in Table 4a. SeqTopK introduces only a marginal pre-training overhead compared to vanilla TopK, *i.e.*,0.9%, improving efficiency–performance trade-off. We benchmark the OLMoE model's decoding efficiency for online SeqTopK and the overhead introduced by Expert-Cache, using *huggingface.gen()* for fair comparison, with KV-Cache enabled for both methods. As shown in Table 4b, online SeqTopK achieves nearly identical throughput to standard TopK. The additional memory overhead of Expert-Cache is negligible, increasing peak GPU memory usage by only 0.8%. As discussed in Section 3.2, its theoretical memory cost is significantly lower than KV-Cache and it can be integrated with minor modifications.

## 4.4 EMPIRICAL UNDERSTANDINGS

In this section, we further provide in-depth understanding of SeqTopK regarding its adaptive behaviors, the distribution of activated experts, and sensitivity w.r.t. batch size compared to BatchTopK.

**SeqTopK Activates More Experts on Hard Tokens.** Inspired by Wang et al. (2025), we study how token entropy correlates with the average number of activated experts. The token entropy for token $t$ is calculated by: $H_t = -\sum_{j=1}^{V} p_{t,j} log(p_{t,j})$, where $p_t \in \mathbb{R}^V$ is the corresponding probability distribution over the vocabulary. A token with higher entropy indicates greater uncertainty in the model's prediction, so ideally, the model should allocate more resources to process it. As shown in

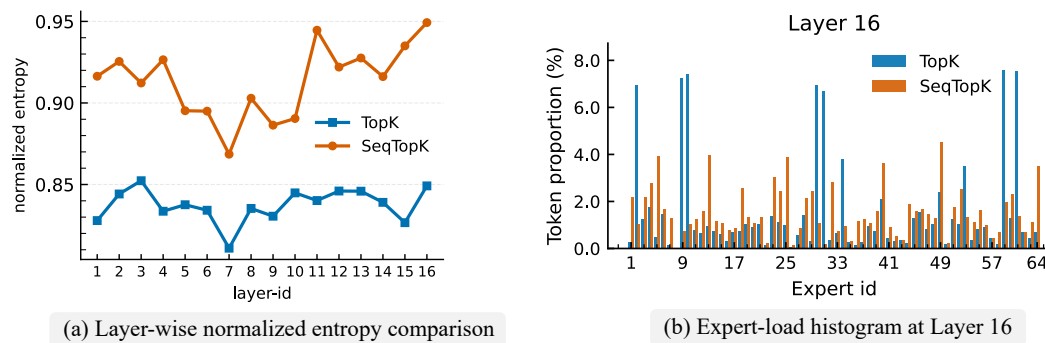

(a) Layer-wise normalized entropy comparison     (b) Expert-load histogram at Layer 16

Figure 4: **Routing Dynamics of SeqTopK.** (a) Layer-wise normalized entropy comparison. *Higher* entropy means more *balanced* expert utilization. SeqTopK consistently exhibits higher entropy than TopK, suggesting that its sequence-level routing encourages more uniformed (*i.e.*, balanced) expert utilization. (b) Expert-load histogram at layer 16. SeqTopK presents smoother and more balanced expert utilization compared to TopK.

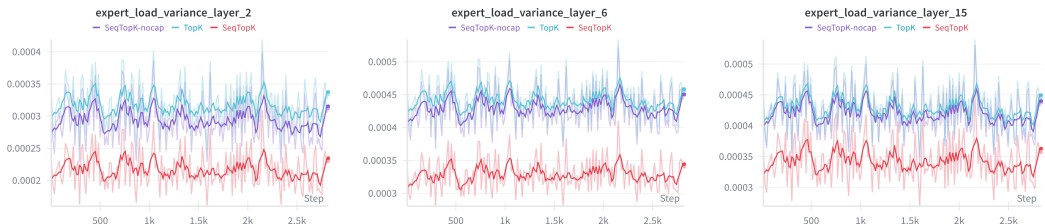

Figure 5: **Expert load variance during training.** Expert load variance was investigated at layers 2, 6, and 15 during OLMoE fine-tuning on GSM8K. SeqTopK with an upper bound consistently delivered the lowest load variance across all tested layers. In comparison, SeqTopK without a cap showed a slightly lower variance than TopK, and crucially, it remains robust without leading to training instability or severe load imbalance.

Figure 3, we sample over 10K tokens generated by a SeqTopK fine-tuned Qwen1.5 and calculate the correlation between token entropy and activated experts. With its context-aware design, SeqTopK distributes the expert budget according to token difficulty: high-entropy tokens, *often those that steer the direction of reasoning*, receive more experts, while low-entropy tokens, *typically carrying out routine steps*, are assigned fewer. Figure 3(b) presents an example of expert allocation. Turning words with high entropy, like "therefore," receive more experts, while low-entropy symbols, like "+" get fewer. Beyond high-entropy tokens, SeqTopK also favors key content words (*e.g.*, factor) and the "$" symbol, which marks the start of a math expression.

**SeqTopK Enables More Balanced Expert Utilization.** We first analyze routing behavior and expert load to compare SeqTopK with standard TopK. Following Lepikhin et al. (2020); Wu et al. (2024a), we compute the normalized routing entropy over the GSM8K dataset. For a MoE layer, let $p_{i,e}$ be the probability of token $i$ being assigned to expert $e$, averaged over the dataset. The overall routing distribution is $p_e = \frac{1}{N} \sum_{i=1}^{N} p_{i,e}$, where $N$ is the total number of tokens. The routing entropy is then defined as $H = -\sum_{e=1}^{E} p_e \log p_e$, and we normalize it by the maximum possible entropy $\log E$. A *higher* entropy means a more *balanced* expert utilization. As shown in Figure 4, SeqTopK generates softer, more distributed assignment probabilities over experts, and achieves higher entropy across layers. By adaptively allocating experts based on token importance, it balances expert utilization across sequences, resulting in a more robust and fine-grained capacity distribution compared to TopK.

**Robustness of Routing Dynamics.** We monitor expert load variance across layers during OLMoE fine-tuning on GSM8K, as illustrated in Figure 5. The results demonstrate that SeqTopK with an upper bound consistently maintains lower load variance across all tested layers. Crucially, even the unconstrained variant exhibits variance slightly lower than the standard TopK baseline. This

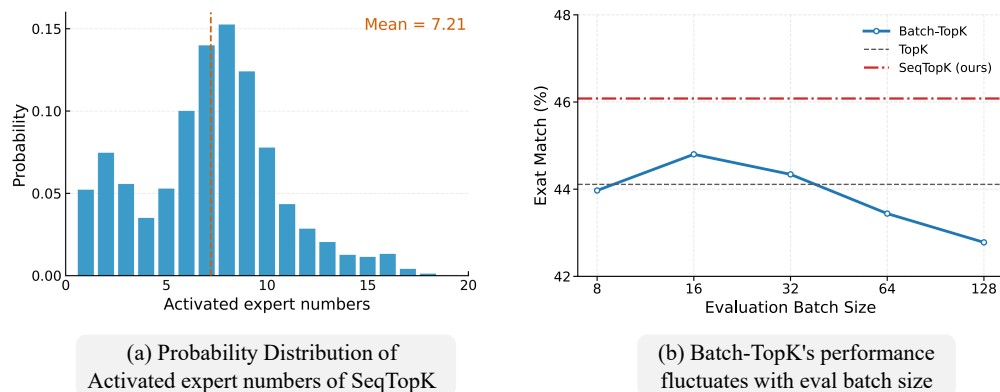

(a) Probability Distribution of
Activated expert numbers of SeqTopK

(b) Batch-TopK's performance
fluctuates with eval batch size

Figure 6: **(a) Expert Activation Pattern of SeqTopK (w/o token-level upper bound).** We present the distribution of OLMoE's expert activation patterns on the GSM8K dataset with the SeqTopK routing strategy. SeqTopK learns to adaptively activate each token with a varying number of tokens from 1 (lower bound) to 18, following a normal-like distribution. **(b) Batch-TopK's performance is highly influenced by the evaluation batch size** and cannot consistently outperform TopK, while SeqTopK could.

empirical evidence confirms that SeqTopK is intrinsically robust, ensuring training stability without relying on rigid per-token caps to prevent load imbalance.

**Normal-like Distribution of Activated Experts.** To further understand the difference between SeqTopK and TopK, we visualize the activated pattern in SeqTopK in Figure 6(a). Given that MoE models consist of multiple MoE layers, we collect data on the number of activated experts per token per layer and show the expert activation patterns. Specifically, we analyze the expert activation distribution for 10K tokens (without token-level upper bound) across 16 layers of the OLMoE model on GSM8K dataset. Compared to a fixed TopK ($K = 8$) selection, SeqTopK dynamically allocates the token budget ranging from 1 to 18, following a normal-like distribution. This demonstrates an emergent adaptiveness, where the model decides the expert budget activation for each token at each layer, resulting in performance gain.

**Sensitivity w.r.t. Batch Size.** We further investigate the effect of evaluation batch size for different methods. Since the total expert budget of BatchTopK is influenced by the training batch size, it is unclear whether this method would stay robust during evaluation. As shown in Figure 6(b), BatchTopK achieves peak accuracy when the training and evaluation batch sizes match (16 per device in our setup) but degrades as the evaluation batch size increases. While effective in MoE-based diffusion models, its sensitivity to batch size limits practicality, where large evaluation batches are common, leading to unstable inference.

## 5 CONCLUSION & DISCUSSION

In this work, we introduced Sequence-level TopK (SeqTopK), a minimal yet powerful replacement for standard TopK routing. By comparing tokens within each sequence under the same overall budget as TopK, SeqTopK enables context-aware allocation of expert capacity: difficult, informative tokens receive more experts, while trivial tokens receive fewer. The method is entirely parameter-free, requires neither thresholds or auxiliary predictors, and can be implemented with only a few lines of code while adding less than 1% overhead.

Several studies have explored *expert-level specialization* via grouping (Guo et al., 2025) or merging (Li et al., 2025b). In contrast, we focus on *token-level specialization*. SeqTopK functions as a horizontal routing mechanism within the FFN, optimizing computational resource on its **width**. This distinct operational scope makes it fully orthogonal to other efficiency paradigms: token pruning (Bolya et al., 2022; Zhang et al.) reduces sequence length, early exiting (Schuster et al., 2022) bypasses layers (depth), and speculative decoding (Leviathan et al., 2023) acts as an external generation strategy. Consequently, SeqTopK is seamlessly compatible with these frameworks, enabling autonomous resource allocation across all dimensions.

**Ethics Statement.** This work complies with the ICLR Code of Ethics. Our research primarily utilizes publicly available datasets and pretrained models, and we do not foresee any direct negative societal impacts or ethical concerns arising from our methodology.

**Reproducibility.** We provide detailed descriptions of our methodology, datasets, model configurations, and evaluation metrics in the main text and Appendix. Upon acceptance, we will release source code and scripts to enable full replication of our experiments.

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

# A    RELATED WORK

**Mixture-of-Experts.** The concept of Mixture-of-Experts (MoE) was first introduced by Jacobs et al. (1991); Jordan & Jacobs (1994) as a way to model heterogeneous data with specialized modules. In this framework, only a subset of parameters, called *experts*, is activated for each input, allowing the model to scale capacity without proportionally increasing computation. Shazeer et al. (2017) extended this idea to large-scale language modeling, training LSTM-based MoE models that achieved performance competitive with dense counterparts. Early MoE works (Roller et al., 2021; Dai et al., 2022) adopt fixed routing strategies, in which each expert is typically assigned a specific role to ensure stable routing and training. Subsequently, GShard (Lepikhin et al., 2020) and Switch Transformer (Fedus et al., 2022) introduced learnable Top-K routing strategies, enabling MoE language models to scale to unprecedented sizes. However, these methods often suffer from load imbalance across experts, requiring auxiliary balancing losses (Wang et al., 2024a). Building on this line, Dai et al. (2024) proposed fine-grained experts that increase the total number of experts by decomposing large experts into smaller ones (e.g., splitting $8$ experts into $64$ experts with $0.25\times$ size each), thereby substantially improving combinatorial flexibility and enabling more precise specialization. Liu et al. (2024a;b) further introduce shared experts to capture and consolidate common knowledge across varying contexts and incorporate an expert-grouping strategy to enhance expert specialization. Beyond advances in model architectures and training strategies, MoE concepts have been broadly adopted in diverse domains, including large language models (Jiang et al., 2024; Muennighoff et al., 2024; Guo et al., 2025; Li et al., 2025a; Yang et al., 2025; Team et al., 2025a), multimodal LLMs (Li et al., 2024; Lin et al., 2024; Li et al., 2025c; Wu et al., 2024b), diffusion models (Balaji et al., 2022; Feng et al., 2023; Sun et al., 2024), and attention mechanisms (Shen et al., 2024; Jin et al., 2024b; Fu et al., 2024; Piękos et al., 2025).

Despite these advances, recent studies suggest that *ultra-sparse* MoE architectures can offer superior efficiency with competitive performance. For example, Qwen3-Next (Yang et al., 2025) introduces an ultra-sparse MoE with only 10 activated experts per token out of 512 total experts, achieving competitive performance with dense models while incurring less than one-tenth of the training cost of dense models. However, our experiments, as shown in Figure 1 (c), show that fixed Top-K routing struggles in such regimes, as the constant number of activated experts per token limits performance under extreme sparsity regime.

To address this limitation, we propose SeqTopK, which allocates each token's expert budget via sequence-level comparison, assigning more experts to high-entropy tokens and fewer to low-entropy ones for adaptive routing in ultra-sparse regimes.

**Adaptive Token Computation.** A central idea in efficient model design is to allocate computation adaptively based on token importance. This principle has been applied from compact representations (Kusupati et al., 2022; Wen et al., 2025) to large language models (Hou et al., 2020; Schuster et al., 2022; Dehghani et al., 2018), consistently showing that selective computation can reduce cost while preserving, or even improving, generalization. Within the MoE framework, several works explore token-level dynamic routing. Huang et al. (2024); Lu et al. (2024) propose Top-P selection, where a token activates experts until their cumulative probability exceeds a threshold $p$. ReMoE (Wang et al., 2024c) replaces TopK and softmax with ReLU gating, coupled with $L_1$ regularization to encourage sparsity and balance. Other methods take a global perspective: Guo et al. (2024); Wu et al. (2025) add a learnable threshold or a judging expert to enforce cross-token adaptivity. Jin et al. (2024a); Team et al. (2025b) add zero-compute experts, allowing tokens routed to them to bypass to the next layer, enabling a dynamic per-token budget. Beyond horizontal expert routing, MoD  (Raposo et al., 2024) and MoR (Bae et al., 2025) has treated adaptive token computation as a depth allocation problem and route tokens vertically to dynamically control computation depth.

Despite these advances, existing approaches face several critical limitations that hinder their widespread adoption. They typically rely on empirical thresholds that generalize poorly across diverse scenarios (Huang et al., 2024; Lu et al., 2024), introduce additional gating parameters that complicate integration with established frameworks (Guo et al., 2024; Wu et al., 2025), and employ multi-stage training procedures that substantially increase computational complexity while undermining training stability (Wang et al., 2024c). Overall, these methods realize dynamic budgeting via *expert reduction*; in contrast, SeqTopK offers a *bidirectional* choice for each token, increasing or decreasing token budgets as needed. SeqTopK neither requires auxiliary parameters nor imposes

additional training overhead, offering a streamlined solution that integrates seamlessly into existing architectures for efficient fine-tuning and continued pre-training.

**Discussion with USMoE.** While the concurrent USMoE (Do et al., 2025) also improve upon the TopK routing, our approaches differ fundamentally. Motivated by **Token Heterogeneity** (Figure 2), SeqTopK dynamically allocates capacity based on token difficulty. In contrast, USMoE targets **Routing Stability** to mitigate representation collapse. Mechanistically, SeqTopK retains the *standard scoring function* but shifts the selection scope from local token-level to **a global sequence-level budget**. USMoE, however, architecturally modifies the scoring formulation by combining token and expert choice scores into a "Unified Score," while **maintaining token-level selection**. Thus, these two works differ fundamentally.

## B    DETAIL EXPERIMENT SETTING

### B.1    PRE-TRAINING EXPERIMENT SETTING

We followed the training pipeline described in Wang et al. (2024c), using The Pile dataset (Gao et al., 2020) for 60,000 steps, which corresponds to approximately 30 billion tokens. This training duration exceeds the compute-optimal dataset size suggested by Krajewski et al. (2024), ensuring the models reach convergence. Our pre-training experiments were conducted at two distinct sparsity levels, activating either 8 out of 64 experts or 4 out of 128 experts for each token. The specific configurations for each model are detailed in Table 5. We used a byte pair encoding (BPE) tokenizer (Sennrich et al., 2015) and maintained consistent training parameters across all models. We employed the AdamW optimizer (Loshchilov & Hutter, 2017) with ZeRO optimization (Loshchilov & Hutter, 2017), using default $\beta_1 = 0.9$ and $\beta_2 = 0.999$ values. The learning rate was set to $5e^{-4}$ with a cosine scheduler and the coefficient for the auxiliary loss (Wang et al., 2024a)is set to $0.01$.

For evaluation, we evaluate the zero-shot performance of the trained models on the following downstream tasks: LAMBADA (Paperno et al., 2016); RACE (Lai et al., 2017); ARC (Clark et al., 2018). All models were trained on 8 NVIDIA A100 GPUs.

Table 5: Configurations for different models.

| Methods | #Parameters | Total Experts | Activated Experts | hidden_size | num_layers | $\lambda_0$ | $\alpha$ |
|---------|-------------|---------------|-------------------|-------------|------------|-------------|----------|
| TopK    | 182M        | 64/128        | 8/4               | 768         | 12         | -           | -        |
| ReLU    | 182M        | 64/128        | 8/4               | 768         | 12         | $1e^{-8}$   | 1.2      |
| SeqTopK | 182M        | 64/128        | 8/4               | 768         | 12         | -           | -        |

### B.2    LLM FINE-TUNING EXPERIMENT SETTING

We select OLMoE-7B-A1B (Muennighoff et al., 2024) and Qwen1.5-14B-A2.7B (Team, 2024) as backbone models and use the codebase provided by Yao et al. (2025) for downstream fine-tuning. For the math domain, we train on GSM8K (Cobbe et al., 2021) and evaluate zero-shot performance. For the code domain, we train the models on the Python subset of the enormous CodeAlpaca dataset (Luo et al., 2023) to simulate a more targeted LLM customization scenario, and evaluate on HumanEval (Chen et al., 2021) and MBPP (Austin et al., 2021). We also incorporate evaluation on the Specialized Tasks proposed by Wang et al. (2024b). All models are optimized within 100 steps under the same training setup, and TopK and SeqTopK use **identical hyperparameters** to ensure a fair comparison; the training hyperparameters for each model–task pair are listed in Table 6. All models were trained on 8 NVIDIA A100 GPUs.

We evaluate zero-shot performance on GSM8K (Exact Match) and HumanEval (Pass@1), 3-shot performance on MBPP (Pass@1), and use GPT-4o (Hurst et al., 2024) to assess the quality of answers on summarization and legal tasks following Wang et al. (2024b).

### B.3    DISCUSSION ON SENTENCE SEGMENTATION STRATEGY

The choice of segmentation strategy (e.g., fixed-size chunks, syntactic boundaries, or dynamic criteria) is vital for efficient large-scale training on long documents. This often involves packing samples

Table 6: Configurations for fine-tuning MoE models

| Model | Dataset | Epoch | Batch_Size | Lr | Warmup_Ratio | Gradient_Ckecpointing |
|-------|---------|-------|-----------|-----|--------------|----------------------|
| OLMOE | GSM8K | 3 | 256 | 1e-6 | 0.03 | False |
| | Codealpha | 2 | 256 | 1e-6 | 0.03 | False |
| | Summary | 2 | 256 | 2e-5 | 0.03 | False |
| | Law | 4 | 256 | 2e-5 | 0.03 | False |
| Qwen1.5 | GSM8K | 4 | 384 | 1e-6 | 0.1 | True |
| | Codealpha | 2 | 256 | 2e-6 | 0.1 | True |
| | Summary | 1 | 256 | 1e-6 | 0.1 | True |
| | Law | 4 | 128 | 1e-5 | 0.1 | True |

into one sequence for efficiency (Raffel et al., 2020), which necessitates methods like the 4D attention mask proposed by S. (2024) to confine attention to individual sequences. SeqTopk could also leverage these techniques; for instance, a 4D mask could enforce syntactic boundaries for its context-level routing, keeping it within a single, semantically-consistent sentence.

## C  TRAINING-FREE SEQTOPK

We investigate the effect of incorporating the online SeqTopK decoding strategy into a fine-tuned TopK model without additional training. As shown in Table 7, the training-free SeqTopK variant with Expert Cache achieves performance comparable to TopK, highlighting its seamless compatibility with modern MoE frameworks. To fully exploit SeqTopK's potential, further fine-tuning (fewer than 100 steps) is required to enable the model to autonomously utilize context awareness for expert routing, as demonstrated by the results in Table 1.

Table 7: **Results of *Training-Free* SeqTopK on top of TopK fine-tuned model.**

| *OLMoE-A1B-7B* Routing Methods | K | Sparsity Ratio | GSM8k 0-shot EM | MBPP 3-shot Pass@1 | HumanEval 0-shot Pass@1 | Summary 0-shot Score | Law 0-shot Score | Avg - Score |
|---------------------------------|---|---------------|------------------|---------------------|--------------------------|----------------------|-------------------|-------------|
| Base | 8 | 1/8 | 15.58 | 19.80 | 10.97 | 7.49 | 5.70 | 11.91 |
| TopK | 8 | 1/8 | 44.11 | 21.04 | 13.41 | 45.31 | 24.89 | 29.74 |
| SeqTopK | | | 44.04 | 20.58 | 13.03 | 44.87 | 24.17 | 29.34 |

## D  ABLATIONS OF BOUND SELECTION

Table 8 presents an ablation study on the selection of the upper bound for SeqTopK, evaluating configurations ranging from $K+1$ to $K+4$ and an unbounded setting across 5 benchmark datasets. We find that while SeqTopK's performance exhibits slight fluctuations depending on the specific upper bound chosen. For example, $K+3$ achieves the highest score on the HuamnEval dataset. Despite these fluctuations, SeqTopK consistently delivers robust performance improvement under most of the evaluated settings. As for the selection of lower bound, our core motivation, as discussed in Figure 2(a), is that not all tokens require the same computational resources. By setting a high lower bound (e.g., $K/2$), we force the model to over-allocate computation, artificially *narrows the model's adaptive flexibility* (*i.e.*,search space) and, as the Table 8 shows, results in a performance drop.

## E  THEORETICAL INSIGHT

In this section, we provide theoretical insights into the superiority of SeqTopK over TopK by framing the routing mechanism as a *constrained resource allocation problem*. We posit that the core

Table 8: **Ablations of bound selection of SeqTopK**. We present an ablation study on the selection of the upper bound for SeqTopK, ranging from $K+1$ to $K+4$, as well as configurations with no upper bound and $K/2$ as lower bound.

| *OLMoE-A1B-7B* Routing Methods | K | Lower bound | Upper bound | GSM8k MBPP 0-shot EM | HumanEval 3-shot Pass@1 | 0-shot Pass@1 |
|---|---|---|---|---|---|---|
| Base TopK | 8 | K | K | 15.58 44.11 | 19.80 21.04 | 10.97 13.41 |
| **SeqTopK** | 8 | 1 | K+1 | 46.07 | 22.38 | 14.63 |
| | | | K+2 | **46.09** | **23.21** | **15.24** |
| | | | K+3 | 45.15 | 22.51 | **15.24** |
| | | | K+4 | 44.54 | 22.00 | 15.01 |
| | | | - | 44.09 | 21.80 | 14.02 |
| | | K/2 | K+2 | 44.38 | 22.21 | 14.37 |

advantage of SeqTopK lies in its ability to exploit the inherent **heterogeneity** of input instances. By modeling the number of activated features (experts) as a limited computational budget, we demonstrate that an adaptive strategy (corresponding to SeqTopK) achieves optimality by adhering to the *equimarginal principle (Marshall, 2013)*, dynamically shifting resources from "easy" samples to "hard" samples.

### E.1 PRELIMINARIES

Consider a dataset $\mathcal{D} = \{(x_i, y_i)\}_{i=1}^{N}$ with $N$ data points. To gain theoretical insight without loss of generality, we analyze a simplified setting where the *resource is modeled as the continuous number of features* utilized for each instance. Let $m_i \in \mathbb{R}_{\geq 0}$ denote the number of features allocated to the $i$-th data point and $\ell_i(m_i)$ represent the loss function for the $i$-th instance given $m_i$ features (e.g., the negative log-likelihood in logistic regression). The total feature budget is fixed at $B$. We assume a continuous relaxation of $m_i$ for gradient-based analysis. Following Marshall (2013), we establish the standard assumption of the principle of diminishing marginal returns in Assumption 1. Based on this, we define two resource allocation strategies that satisfy the budget constraint $\sum_{i=1}^{N} m_i = B$: **uniform** (i.e., TopK routing) and **adaptive** (i.e., SeqTopK routing) allocation.

**Assumption 1** (Monotonicity and Convexity (Marshall, 2013)). *For every instance $i$, the loss function $\ell_i(m_i)$ is strictly decreasing and strictly convex with respect to the feature count $m_i$. That is:*

$$\ell_i'(m_i) < 0, \quad \ell_i''(m_i) > 0, \quad \forall m_i > 0. \tag{11}$$

*This reflects the principle of diminishing marginal returns: adding features reduces loss, but the benefit per feature decreases as the total number of features increases.*

**Definition 1** (Uniform Allocation). *The Uniform strategy assigns an equal feature budget $\bar{m}$ to all instances:*

$$m_i^{uni} = \bar{m} = \frac{B}{N}, \quad \forall i. \tag{12}$$

*The total loss is denoted as $\mathcal{L}_{uni} = \sum_{i=1}^{N} \ell_i(\bar{m})$.*

**Definition 2** (Adaptive Allocation). *The Adaptive strategy seeks the optimal allocation $\{m_i^*\}_{i=1}^{N}$ that minimizes the total loss:*

$$\min_{\{m_i\}} \sum_{i=1}^{N} \ell_i(m_i) \quad s.t. \quad \sum_{i=1}^{N} m_i = B. \tag{13}$$

*The total loss is denoted as $\mathcal{L}_{adapt} = \sum_{i=1}^{N} \ell_i(m_i^*)$.*

### E.2 OPTIMAL STRATEGY

**Theorem 1.** *Under Assumption 1, the adaptive strategy strictly outperforms uniform strategy ($\mathcal{L}_{adapt} \leq \mathcal{L}_{uni}$). Furthermore, the magnitude of the loss reduction is proportional to the variance in the marginal returns among the instances when resources are allocated uniformly.*

*Proof.* We formulate the constrained optimization problem in Equation 13 using the Lagrangian function, with $\lambda$ serving as the Lagrange multiplier for the budget constraint.

$$\mathcal{L}(\boldsymbol{m}, \lambda) = \sum_{i=1}^{N} \ell_i(m_i) + \lambda \left( \sum_{i=1}^{N} m_i - B \right). \tag{14}$$

The first-order optimality conditions, specifically the Karush-Kuhn-Tucker (KKT) conditions, for the resource allocation problem directly lead to the Equimarginal Principle (Marshall, 2013).

$$\frac{\partial \ell_i}{\partial m_i} + \lambda = 0 \implies \ell_i'(m_i^*) = -\lambda, \quad \forall i. \tag{15}$$

This implies that at the optimal allocation $m^*$, the marginal gradient is **identical** across all instances. Let this optimal gradient be $g^* = -\lambda$. To quantify the gap $\Delta \mathcal{L} = \mathcal{L}_{uni} - \mathcal{L}_{adapt}$, we perform a second-order Taylor expansion of $\ell_i(m_i^*)$ around the uniform point $\bar{m}$:

$$\ell_i(m_i^*) \approx \ell_i(\bar{m}) + \ell_i'(\bar{m})(m_i^* - \bar{m}) + \frac{1}{2}\ell_i''(\bar{m})(m_i^* - \bar{m})^2. \tag{16}$$

Summing over all $i$, and noting that $\mathcal{L}_{adapt} = \sum \ell_i(m_i^*)$ and $\mathcal{L}_{uni} = \sum \ell_i(\bar{m})$:

$$\mathcal{L}_{adapt} \approx \mathcal{L}_{uni} + \sum_{i=1}^{N} \ell_i'(\bar{m})\Delta m_i + \frac{1}{2}\sum_{i=1}^{N} \ell_i''(\bar{m})(\Delta m_i)^2, \tag{17}$$

where $\Delta m_i = m_i^* - \bar{m}$. From the budget constraint, we know $\sum \Delta m_i = 0$. However, the first-order term $\sum \ell_i'(\bar{m})\Delta m_i$ does not vanish because gradients $\ell_i'(\bar{m})$ are **heterogeneous**. This heterogeneity implies that not all data (tokens) require the same resource (expert budget), a conclusion empirically motivated in Figure 2 and discussed in Section 3.1. To relate $\Delta m_i$ to the gradients, we linearize the gradient condition. Near $\bar{m}$, we approximate:

$$\ell_i'(m_i^*) \approx \ell_i'(\bar{m}) + \ell_i''(\bar{m})\Delta m_i. \tag{18}$$

Since optimality requires $\ell_i'(m_i^*) = g^*$ (constant for all $i$):

$$g^* \approx \ell_i'(\bar{m}) + \ell_i''(\bar{m})\Delta m_i \implies \Delta m_i \approx -\frac{\ell_i'(\bar{m}) - g^*}{\ell_i''(\bar{m})}. \tag{19}$$

For analytical insight, assume a constant local curvature (Hessian) $\ell_i''(\bar{m}) \approx H > 0$ across samples. Then $\Delta m_i \approx -\frac{1}{H}(\ell_i'(\bar{m}) - g^*)$. Substituting this back into the loss expansion:

$$\mathcal{L}_{adapt} - \mathcal{L}_{uni} \approx \sum_{i=1}^{N} \ell_i'(\bar{m}) \left[ -\frac{1}{H}(\ell_i'(\bar{m}) - g^*) \right] + \text{h.o.t.} \tag{20}$$

$$\approx -\frac{1}{H} \sum_{i=1}^{N} \left( \ell_i'(\bar{m})^2 - \ell_i'(\bar{m})g^* \right). \tag{21}$$

Since $\sum \Delta m_i = 0$, it implies $\sum(\ell_i'(\bar{m}) - g^*) = 0$, so $g^*$ is essentially the mean gradient $\bar{g} = \frac{1}{N}\sum \ell_i'(\bar{m})$. The expression simplifies to:

$$\mathcal{L}_{uni} - \mathcal{L}_{adapt} \approx \frac{1}{2H} \sum_{i=1}^{N} (\ell_i'(\bar{m}) - \bar{g})^2 \geq 0. \tag{22}$$

$\square$

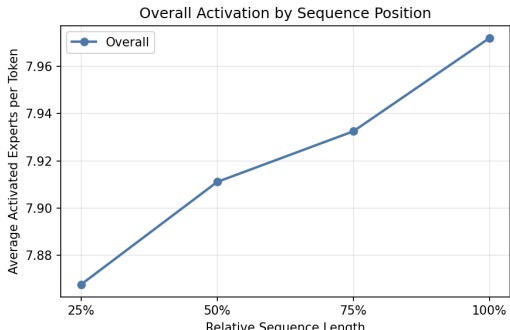

Figure 7: **Online SeqTopK activation patterns on GSM8K.** The average number of activated experts *increases* with sequence length. This upward trend confirms that the model adaptively reallocates resources, shifting computational capacity from early, context-poor tokens to later, context-rich ones to optimize expert utilization.

## F  ONLINE SEQTOPK ACTIVATION PATTERNS

In this section, we investigate whether the cumulative budget constraint in SeqTopK leads to resource under-utilization compared to standard TopK. Empirical results suggest the contrary. As illustrated in Figure 7, the average number of activated experts *increases* with sequence length, regardless of answer correctness. This trend indicates that the flexibility of SeqTopK introduces an implicit *banking mechanism* that aligns computational expenditure with semantic density. Specifically, the model minimizes expert usage for initial, low-entropy tokens to accumulate a computational surplus. This surplus is subsequently expended on later tokens where semantic ambiguity is higher. Notably, we observe a distinct activation surge during the later stages of incorrect responses. This phenomenon further validates that SeqTopK prioritizes high-entropy tokens, allocating maximal resources (Figure 3) to uncertain contexts where the model is most prone to generation failure. Consequently, the fixed budget is not left unused; rather, the model optimizes its allocation across the temporal dimension of the sequence.

## G  COMPARISON WITH MOE++

We have added a comparison between MoE++ in a fine-tuning setting and found that SeqTopK outperforms MoE++ by a significant margin (**46.09** vs. 18.91). Following Jin et al. (2024a), we implemented MoE++ on top of the pre-trained OLMoE model by adding one "zero" and one "copy" expert, and then fine-tuned the model under the exact same hyperparameters used for SeqTopK. As highlighted in Table 9, SeqTopK establishes a substantial **+143%** performance advantage over MoE++. While MoE++ yields a marginal improvement over the base model (18.91 vs. 15.58), it is hindered by **fundamental architectural incompatibilities**, which means it is *only designed for pre-training from scratch.* In contrast, SeqTopK integrates seamlessly with existing pre-trained checkpoints, delivering large gains without re-training from scratch.

Table 9: **Comparison with MoE++ (fine-tuned) on the GSM8K benchmark.**

| Methods | GSM8K |
|---------|-------|
| Base | 15.58 |
| TopK | 44.11 |
| MoE++ | 18.91 |
| SeqTopK | **46.09** |

## H  SENSITIVITY WITH SEQUENCE LENGTH

In this section, we analyzed the model's robustness to varying sequence lengths during inference. We observe that SeqTopK **does not introduce additional fragility** to length variations. As Table 10 shows, SeqTopK consistently outperforms the Top-K baseline across short, medium, and long contexts (e.g., +2.8% for lengths [100,250)). While accuracy for both methods naturally declines as length increases (reflecting increased reasoning complexity), SeqTopK's degradation trend *closely tracks the baseline*, indicates that our dynamic routing mechanism is robust to length variations.

The observed sensitivity is attributable to the intrinsic limits of the pre-trained model rather than the routing strategy.

Table 10: **Performance vs. Inference Answer Length of fine-tined OLMoE on GSM8K**

| Answer length | [0,50) | [50,100) | [100,150) | [150,250) | [250,) |
|---|---|---|---|---|---|
| TopK | 60.48 | 51.78 | 33.71 | 25.08 | 0.0 |
| SeqTopK | 63.72 | 52.99 | 35.41 | 25.04 | 0.0 |

## I    MORE VISUALIZATION

To investigate how the model handles ambiguity, we visualized the routing probability distribution for tokens that require more than $K$ experts (Figure 8). As shown in Figure 8(a), TopK exhibits a rigid, high-variance distribution with dominant spikes reaching $0.14$; this behavior is suboptimal for tokens that semantically align with multiple experts SeqTopK addresses this by enabling a smoother probability landscape. As illustrated in Figure 8(b), SeqTopK suppresses the sharp peaks observed in TopK (reducing the maximum probability from $0.14$ to $\approx 0.08$) and allocates more weight to secondary experts.

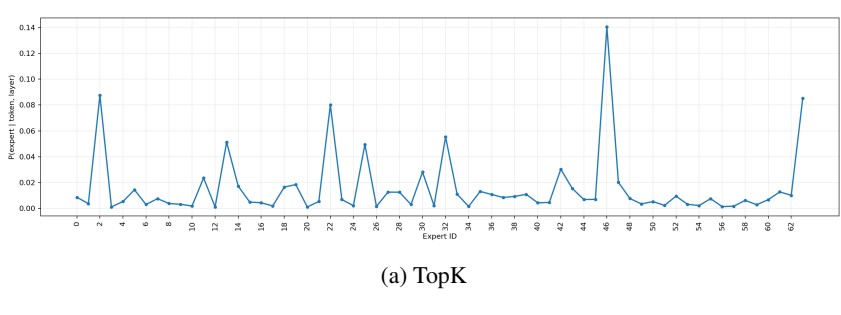

(a) TopK

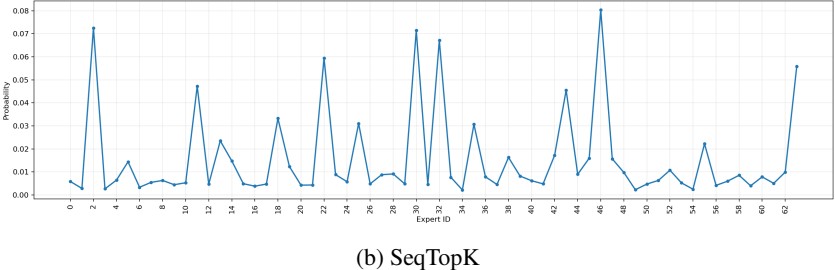

(b) SeqTopK

Figure 8: **Impact of SeqTopK on Routing Distribution of token with $> K$ experts.** (a) **TopK** results in a highly peaked distribution where the dominant expert receives a probability of $\approx 0.14$. (b) **SeqTopK** enables a softer distribution for tokens requiring $> K$ experts, flattening the peak probability to $\approx 0.08$ and spreading weights more broadly across potential experts.

## J    LIMITATIONS

In this section, we discuss the boundaries and limitations of SeqTopK. First, the method is currently specialized for sparse MoE routing, which means it is inapplicable to dense architectures. However, the core principle of *adaptive budget allocation* is generalizable to other domains, such as dynamic attention pruning, which we leave it for future work. Second, SeqTopK introduces only a slight overhead: as shown in Table 4b, throughput decreases by $\approx 1\%$ (141.23 vs. 139.41 tokens/s), with marginal increases in memory and training time ($< 1\%$). We argue that this minimal cost is a highly justified trade-off given the substantial performance gains of +5.9% and +3.6% demonstrated in Tables 1 and 2, respectively.

## K  LLM USAGE STATEMENT.

In line with the ICLR policy, we disclose the use of Large Language Models during the preparation of this manuscript. Our use of these tools was strictly limited to assistance with language and formatting. Specifically, we employed an LLM to correct grammatical errors and improve the clarity and readability of sentences. The LLM had no role in the core scientific aspects of this work, including research ideation, methodological design, experimental analysis, or the generation of any results or conclusions. All intellectual contributions and the core content of this paper are solely the work of the authors.

