# OpenReview forum: "Route Experts by Sequence, Not by Token"
_ICLR.cc/2026/Conference — Submitted to ICLR 2026_

### Official Review · Reviewer_atcY · 2025-10-30

**Soundness:** 1
**Presentation:** 3
**Contribution:** 1
**Rating:** 2
**Confidence:** 4

**Summary:**

The paper proposes an adaptive routing strategy for sparse mixture-of-experts (MoE) models called Sequence-level Top K (SeqTopK), which replaces the conventional per-token TopK based expert activation budget with a per-sequence budget. Instead of selecting a fixed number of experts for each token, SeqTopK computes a total budget of T×K experts (where T is the number of tokens) and allocates that budget dynamically across all tokens in the sequence - effectively giving more experts to “harder” tokens and fewer to “easier” ones. During training, it is very lightweight and adds only few lines of code with minimal (1%) overhead.

**Strengths:**

- Clear and well-motivated problem formulation: Identifies that token-level Top-K routing misallocates compute - over-serving easy tokens and under-serving difficult ones.
- Innovative sequence-level budgeting: SeqTopK enables context-aware expert allocation by considering each token’s relative importance within the sequence, not just its standalone score. But its working during inference is not accurate.

- Practical compute bound: Introduces a simple yet effective constraint of minimum and maximum number of experts being selected for each token [1, K+2] to maintain control over compute for all tokens.

**Weaknesses:**

Online SeqTopK:

- I see a major flaw with the approach proposed for autoregressive decoding during inference. The method is sound during training when we have access to all tokens expert's scores in a sequence. But the methodology in section 3.2 for online SeqTopK is not correct in providing flexible K for different tokens.

- The method proposes to limit expert selection budget to $m.K$ for every sequence index $m$ where $1<m<T$. But we can not change the past expert selection at any point in time of generation.
- Lets say we start at seq index $m=1$ then we will select $1.K=K$ experts for current token. Next, we move to generate second token $m=2$, then our budget is to select total $2.K$ experts till $m=2$ sequence index. But since we have already selected $K$ experts for the previous token, we are left to select $K$ experts for the current $m=2$ token as well. And in the same fashion, the method continues.
- The proposed online SeqTopK effectively collapses to the standard token-level TopK routing as the expert selection budget at each step $m$ cannot retroactively adjust past allocations, each new token still receives exactly $K$ experts. Thus, in an autoregressive setting, the online variant provides no real deviation from per-token TopK behavior.
- There is no experimental results showing otherwise for the online set up. There is one section C in appendix including results of online SeqTopK and it shows similar results as regular TopK.

Other parts:
 - Dependence on sequence length: Unclear how SeqTopK scales to models with different or longer context lengths, especially when training and inference sequence lengths differ.
- Expert allocation bound ([1, K+2]): The rationale for this specific range is unclear—should the lower bound also depend on K, and does it always guarantee exactly T×K experts per sequence?
- Lack of expert score analysis: No detailed study on how sequence-level constraints reshape router distributions or whether fine-tuning affects only the router or all parameters; unclear how tokens with >K experts adjust their distributions.

**Questions:**

I recommend the authors develop a practical online inference strategy so SeqTopK can adapt expert allocation during autoregressive generation rather than degenerating to per-token TopK.

---

> ### Author Response · Authors · 2025-11-21
>
> Thank you very much for your careful review. We respectfully note that there may be some misunderstandings regarding the core contribution of our work -- specifically, the design and role of Online SeqTopK. To clarify this important aspect, we provide a more detailed explanation below and address each of your concerns point by point.
>
> ---
> **Q1:** Concerns about Online SeqTopK.
>
> > - I see a major flaw with the approach proposed for autoregressive decoding during inference. The method is sound during training when we have access to all tokens expert's scores in a sequence. But the methodology in section 3.2 for online SeqTopK is not correct in providing flexible K for different tokens.
> > - Lets say we start at seq index  $m=1$ then we will select  $1 \cdot K=K$ experts for current token. Next, we move to generate second token $m=2$  , then our budget is to select total  $2 \cdot K=2K$ experts till  $m=2$ sequence index. But since we have already selected  $K$ experts for the previous token, we are left to select  $K$ experts for the current  $m=2$ token as well. And in the same fashion, the method continues.
>
> **A1.1**: We are afraid that there are some misunderstandings here. The key is that we **do not change the past**; we only use the _scores_ from past tokens (in the Expert cache) to set a **dynamic global threshold** for the _current_ token.
>
> Let's walk through the toy example you provided, but with the correct online logic.
> - Let say, there are $N=4$ experts and we will select $K=2$ experts for each token. There are two tokens, where their token-expert weights are $s_1=[0.1,0.2,0.3,0.4]$ and $s_2=[0.8,0.05,0.1,0.05]$.
> - At Seq index $m=1$, we will select Top $1*K =2$ values, and the threshold is **0.3.** And we will store the original score vector in Expert-Cache. Now, token 1's computation is finished and cannot be changed.
> - Moving to the $m=2$ token, where we select Top $2*K=4$ values of $$ [s_1,s_2]=\pmatrix{ 0.1 & 0.2 & 0.3 & 0.4 \cr 0.8 & 0.05 & 0.1 &0.05 } $$
> - The **new dynamic threshold** is the 4th highest score: **0.2**. So the second token is **only assigned to one experts**. Now, token 2's computation is finished and cannot be changed. We add the original score vector $s_2$ into Expert-Cache.
> - As a result, Online Seqtopk: token 1 (2 experts), **token 2 (1 expert)** vs TopK: token 1 (2 experts), **token 2(2 experts)**.  This flexibility allows our model to dynamically *save* computational budget on simpler tokens (like Token 2) and allocate it to more complex ones (e.g., a hypothetical Token 3 might get 3 experts).
> - Crucially, this flexibility does not add computational risk. As stated in **line 227-229**, online SeqTopK is strictly bounded by the T ·K experts of the vanilla formulation and guarantees that for any 1 ≤ m ≤ T , the cumulative number of activated experts **never exceeds** m·K.
>
> >There is no experimental results showing otherwise for the online set up. There is one section C in appendix including results of online SeqTopK and it shows similar results as regular TopK.
>
> **A1.2**:  We are afraid this interpretation of Appendix C is misleading. The primary goal of that section was to investigate a specific _ablation_: applying SeqTopK **on-the-fly** to a router that was **pre-trained only with static TopK**, without any further training.
>
> - **The Meaning of Appendix C:** The similar results in Appendix C simply confirm that a router trained for static constraints cannot spontaneously utilize dynamic budgeting without adaptation. The router **must be fine-tuned** to learn the "Banking Mechanism" (saving budget on easy tokens to spend on hard ones).
> - **The Actual Evidence (Tables 1 & 2):** The **true experimental validation** of Online SeqTopK is presented in the main paper (**Table 1** and **Table 2**). In these experiments, where the model is properly fine-tuned to utilize the mechanism, Online SeqTopK achieves substantial gains (**+5.9%** on GSM8K and **+3.6%** on MBPP).
>
> We hope this detailed toy example and further clarification will resolve your concerns regarding the online SeqTopK mechanism, and we welcome any further discussion on this point.
>
> (continue below)

---

> > ### Author Response · Authors · 2025-11-21
> >
> > **Q2**: Dependence on sequence length: Unclear how SeqTopK scales to models with different or longer context lengths, especially when training and inference sequence lengths differ.
> >
> > **A2**: As you suggested, we analyzed the model's robustness to varying sequence lengths during inference. We observe that SeqTopK **does not introduce additional fragility** to length variations.
> >
> > *Table 1 .* Performance vs. Inference Answer Length (OLMoE on GSM8K)
> >
> > | Answer length |  [0,50)   | [50,100)  | [100,150) | [150,250) | [250,) |
> > | :-----------: | :-------: | :-------: | :-------: | :-------: | :----: |
> > |   TopK Acc    |   60.48   |   51.78   |   33.71   |   25.08   |  0.0   |
> > |  SeqTopK Acc  | **63.72** | **52.99** | **35.41** |   25.04   |  0.0   |
> >
> > As *Table 1* shows, SeqTopK **consistently outperforms** the Top-K baseline across short, medium, and long contexts (e.g., **+2.8%** for lengths [100,250)).
> > While accuracy for both methods naturally declines as length increases (reflecting increased reasoning complexity), SeqTopK's degradation trend **closely tracks** the baseline, indicates that our dynamic routing mechanism is robust to length variations. The observed sensitivity is attributable to the intrinsic *limits of the pre-trained* model rather than the routing strategy.
> >
> > We note that evaluating sensitivity to _training_ length variations requires pre-training from scratch on varying context windows. Due to the limited rebuttal window and prohibitive cost (approx. **10 days on 8x A100s** for one run) , we focus our analysis on **inference robustness**, evaluating how the model behaves across different output lengths *within* its effective operating range. We've included this discussion in **Appendix H**.
> >
> > ---
> > **Q3**: Expert allocation bound ([1, K+2]): The rationale for this specific range is unclear—should the lower bound also depend on K, and does it always guarantee exactly T×K experts per sequence?
> >
> > >Expert allocation bound ([1, K+2]): The rationale for this specific range is unclear—should the lower bound also depend on K?
> >
> > **A3.1:** We sweep the upper bound from *K+1* to *no cap* and find that **$K+2$ provides the optimal trade-off** between flexibility and regularity across varying benchmarks. The results of ablation is presented in *Table 1* below:
> >
> > *Table 1 .* Ablations of bound selection of SeqTopK on OLMoE.
> >
> > | Methods | Lower<br>Bound | Upper Bound | GSM8K     | MBPP      | HumanEval |
> > | :-----: | :------------: | :---------: | --------- | --------- | --------- |
> > |  TopK   |       K        |      K      | 44.11     | 21.04     | 13.41     |
> > | SeqTopK |       1        |     K+1     | 46.07     | 22.38     | 14.63     |
> > | SeqTopK |       1        |     K+2     | **46.09** | **23.21** | **15.24** |
> > | SeqTopK |       1        |     K+3     | 45.15     | 22.51     | **15.24** |
> > | SeqTopK |       1        |     K+4     | 44.54     | 22.00     | 15.01     |
> > | SeqTopK |       1        |      -      | 44.09     | 21.80     | 14.02     |
> > | SeqTopK |      K/2       |     K+2     | 44.38     | 22.21     | 14.37     |
> >
> > - **Upper Bound:** We find that while SeqTopK’s performance exhibits slight fluctuations(e.g., 14.02 to 15.24 on HumanEval) depending on the specific upper bound chosen, it consistently delivers **robust performance improvement** in most cases. Based on these empirical results, we recommend **$K+2$** as a robust default that yields significant gains without the need for task-specific tuning.
> > - **Lower Bound:** Our core motivation, as discussed in **Figure 2(a)**, is that **not all tokens require the same computational resources**. By setting a **high lower bound** (e.g., $K/2$), we _force_ the model to over-allocate computation,  artificially **narrows the model's adaptive flexibility** (*i.e.*,search space) and, as the *Table 1* shows, results in a performance drop.
> >
> > We have included this ablation study and sensitivity analysis in **Appendix D** of the revised manuscript to provide transparency on hyperparameter selection.
> >
> > >Does it always guarantee exactly T×K experts per sequence?
> >
> > **A3.2:**  No, it does not guarantee **exactly** T×K experts per sequence. Online SeqTopK guarantees that for any $1 \le m \le T$, the cumulative number of activated experts **never exceeds** $m \cdot K$,  as stated in **line 227-229**. This ensures our method is always _bounded_ by the same total $T \cdot K$ budget as the vanilla TopK formulation, but it is not forced to _use_ that full budget.
> >
> > In fact, we also shows that the mean number of activated experts per token is **7.21**, which is _less_ than the standard $K=8$. in **Figure 6(a)**. This demonstrates that our model learns to save computation on simpler tokens while always remaining safely within the $T \cdot K$ computational bound.
> >
> > (continue below)

---

> > > ### Author Response · Authors · 2025-11-21
> > >
> > > **Q4**: Lack of expert score analysis: No detailed study on how sequence-level constraints reshape router distributions or whether fine-tuning affects only the router or all parameters; unclear how tokens with >K experts adjust their distributions.
> > >
> > > >No detailed study on how sequence-level constraints reshape router distributions.
> > >
> > > **A4.1:**  Follow your suggestion, we have analyzed how sequence-level constraints reshape router distributions and found that SeqTopK promotes a **more uniform utilization of the expert pool.**  We verified this point from both visual and quantitative perspective.
> > >
> > > - **Visual Evidence (Expert-load Histogram):** As visualized in the **Figure 4(b)**, the standard Top-K baseline suffers from a "spiky" distribution, indicating severe over-reliance on specific experts. In contrast, SeqTopK enables a **visibly smoother and more equitable expert load distribution**.
> > > - **Quantitative Evidence (Expert-load Entropy):** SeqTopK consistently yields **higher Expert-Routing Entropy** compared to the baseline as shown in **Figure4(a)**. This higher entropy quantitatively proves that the model is utilizing the expert pool more broadly.
> > >
> > > By allowing "hard" tokens to access _more_ experts while restricting "easy" tokens to _fewer_, SeqTopK naturally **relieves congestion** on popular experts. We have integrated this detailed analysis into **Section 4.4** of the revised manuscript.
> > >
> > > >Unclear how tokens with >K experts adjust their distributions.
> > >
> > > **A4.2:** Following your suggestion, we analyzed the routing distribution for tokens requiring $>K$ experts. As shown in **Figure 8**, the baseline TopK exhibits a sharp spike with probability $\approx 0.14$. In contrast, SeqTopK learns **a softer distribution** (peak $\approx 0.08$), spreading weights more broadly across potential experts. This adaptivity allows SeqTopK to leverage more compute for harder token wheras save for easy ones.
> > >
> > > ---
> > > Thank you once again for your careful reading and constructive suggestions. We have thoroughly refined the manuscript in accordance with your feedback and have addressed each of your concerns above. We respectfully hope that the updated analyses and results will allow you to re-evaluate our work. Should any questions remain, we would be more than happy to provide further clarification.

---

### Official Review · Reviewer_5nh4 · 2025-11-01

**Soundness:** 2
**Presentation:** 2
**Contribution:** 2
**Rating:** 4
**Confidence:** 4

**Summary:**

The paper proposes a context-level MoE routing strategy that assigns experts based on sequences rather than individual tokens. The authors argue that token-level routing, which assigns a fixed number of experts per token regardless of its informativeness, is suboptimal, and that aggregating routing signals across a context window can lead to more coherent and efficient expert utilization. To implement this, the paper introduces the SeqTopK mechanism, including both an offline variant and an online variant designed for non-causal settings. The authors conduct experiments across multiple tasks and benchmarks to demonstrate the performance benefits of their approach, and provide visualizations to analyze expert usage patterns.

**Strengths:**

The paper presents a clear motivation and introduces the idea of context-level expert routing in a straightforward manner. The writing is concise and easy to follow, and the overall method is described with sufficient clarity. Experiments are relatively comprehensive, covering multiple tasks and including both quantitative results and visualizations to support the analysis.

**Weaknesses:**

* **Sequence Segmentation Strategy.**: The paper promotes context-level expert routing as a key improvement over token-level MoE, yet it fails to specify how the input sequence is segmented for routing purposes. It remains unclear whether fixed-size chunks, syntactic boundaries, or dynamic criteria are used to define each sequence. In long documents, inappropriate or static segmentation may lead to semantic fragmentation or boundary misalignment, ultimately impairing the coherence and effectiveness of expert selection. Ironically, this may negate the intended advantages over token-level granularity, especially considering that token representations already encode contextual information.

* **Inconsistency Between Motivation and Visualization**. The paper argues that assigning each token a fixed number of experts (K), regardless of its informativeness, is suboptimal—a point with which I strongly agree. As noted by the authors in Lines 54–58, *“some tokens are trivial and require little capacity (e.g., 'the')”*. However, this intuition appears to be contradicted by the empirical evidence in Figure 3(b), which shows that "the" is among the tokens associated with the highest number of experts. To reconcile this discrepancy and substantiate the paper’s motivation, more thorough and quantitative analyses are needed.

* **Limitations of Non-Causal Design**. While I appreciate the motivation behind the Online SeqTopK mechanism and agree that it offers a reasonable solution for non-causal settings, there are two potential concerns that warrant further discussion.
  * (1) The constraint that, at step m, the total number of activated experts must not exceed m·K may significantly limit the flexibility of the routing mechanism—especially in long sequences where semantic dynamics can vary drastically. This rigid upper bound could lead to under-utilization of expert capacity in later tokens, particularly when earlier tokens require fewer experts.
  * (2) It remains unclear how well SeqTopK integrates with standard decoding techniques used in autoregressive language models, such as speculative decoding or beam search. Given the increasingly widespread use of such strategies for efficient inference, a discussion (or empirical validation) of SeqTopK’s compatibility with these paradigms would strengthen the practical relevance of the proposed method.

**Questions:**

Please refer to Weaknesses.

---

> ### Author Response · Authors · 2025-11-21
>
> Thank you very much for your careful reading and insightful questions. We truly appreciate the time and effort you devoted to evaluating our work. We address your concerns point-by-point as follows.
>
> ---
> **Q1**: Sequence Segmentation Strategy: The paper promotes context-level expert routing as a key improvement over token-level MoE, yet it fails to specify how the input sequence is segmented for routing purposes. It remains unclear whether fixed-size chunks, syntactic boundaries, or dynamic criteria are used to define each sequence. In long documents, inappropriate or static segmentation may lead to semantic fragmentation or boundary misalignment, ultimately impairing the coherence and effectiveness of expert selection. Ironically, this may negate the intended advantages over token-level granularity, especially considering that token representations already encode contextual information.
>
> **A1**: Thank you for this insightful and practical question on sequence segmentation strategy! We will address your concerns based on different training setting:
> - **Fine-tuning Setting:**   We follow the **standard training recipe** for language models, padding samples to a fixed maximum length (e.g., 4096 tokens). Given that our average sample length is **relatively short** ($\approx 512$ tokens), most instances remain intact and semantically complete. For the rare outliers exceeding the context window, we apply **truncation**, retaining the initial segment to preserve the primary context.
> - **Pre-training Setting:** We leverage *Document Concatenation* and *sliding window* techniques provided by Megatron-LM [1]. Specifically, documents are concatenated via `<eod>` tokens into a continuous stream and sliced into fixed-length sequences. **Texts exceeding the context window are spanned across consecutive samples rather than truncated**, ensuring zero token waste. we further employ *Generalized Attention Masks* and *Position ID resetting* to guarantee mathematical independence between packed documents.
>
> We have added this clarification in the _Experiment Setup_ in **Sec 4.1** and **4.2**.
>
> **Discussion on Sentence Segmentation:** We agree that segmentation strategy is vital for efficient large-scale training. The approach often involves packing different samples into a single sequence to improve computational efficiency [2]. [3] introduces a 4D attention mask that confines attention exclusively to individual sequences. SeqTopk could also leverage these techniques; for instance, a 4D mask could enforce syntactic boundaries for its context-level routing, keeping it within a single, semantically-consistent sentence. We have added this detailed discussion in **Appendix B.3**.
>
> [1] https://github.com/NVIDIA/Megatron-LM
>
> [2] Raffel, Colin, et al. "Exploring the limits of transfer learning with a unified text-to-text transformer." _Journal of machine learning research_ 21.140 (2020): 1-67.
>
> [3] https://huggingface.co/blog/poedator/4d-masks
>
> (continue below)

---

> > ### Author Response · Authors · 2025-11-21
> >
> > **Q2**: Inconsistency Between Motivation and Visualization. The paper argues that assigning each token a fixed number of experts (K), regardless of its informativeness, is suboptimal—a point with which I strongly agree. As noted by the authors in Lines 54–58, “some tokens are trivial and require little capacity (e.g., 'the')”. However, this intuition appears to be contradicted by the empirical evidence in Figure 3(b), which shows that "the" is among the tokens associated with the highest number of experts. To reconcile this discrepancy and substantiate the paper’s motivation, more thorough and quantitative analyses are needed.
> >
> > **A2**: We apologize for the confusion in Lines 54-58. What we intended to convey is that **token difficulty depends on its context rather than the token itself.**  This insight—that the _same_ token type requires varying compute based on its surroundings—is precisely what motivates SeqTopK over rigid static routing. We use the specific instances of "the" in **Figure 3(b)** to elaborate on this point and resolve the apparent discrepancy:
> >
> > - **The _1st_ "the" (Sequence Initiation):** This token appears at the **very beginning of the entire answer** with *no prior context.* As stated in [1], initial tokens often possess **high entropy** because they determine the trajectory of the entire sequence. Our method correctly identifies this as a "hard" token and allocates a **high number of experts**.
> > - **The _2nd_ "the" (Mid-Phrase):** This instance occurs within a highly predictable phrase where the semantic context is stable. Here, the token is genuinely "trivial" and carries little new information. Consistent with our motivation, SeqTopK allocates a **below-average** number of experts (significantly lower than the first "the").
> > - **The _3rd_ "the" (Logical Transition):** This token follows the word *"Therefore,"* which signals a critical logical conclusion [1]. The context shifts from description to reasoning, increasing the information density. SeqTopK detects this **semantic shift** and adaptively allocates **more capacity** to handle the complex context following the transition.
> >
> > This analysis demonstrates that **Figure 3(b) validates, rather than contradicts, our motivation.** A standard Top-K model wrongly assumes all "the" tokens are equal. In contrast, SeqTopK correctly identifies that **the same token type can be either hard or trivial depending on its position and context.**
> >
> > To prevent misunderstanding, we have refined the phrasing in Lines 54-58 of the revised manuscript.  Please do let us know if there are other specific parts that you still find skeptical after reading our response.
> >
> > [1] Wang, Shenzhi, et al. "Beyond the 80/20 rule: High-entropy minority tokens drive effective reinforcement learning for llm reasoning." _arXiv preprint arXiv:2506.01939_ (2025).
> >
> > ---
> > **Q3**: The constraint that, at step m, the total number of activated experts must not exceed m·K may significantly limit the flexibility of the routing mechanism—especially in long sequences where semantic dynamics can vary drastically. This rigid upper bound could lead to under-utilization of expert capacity in later tokens, particularly when earlier tokens require fewer experts.
> >
> > >The constraint that, at step m, the total number of activated experts must not exceed m·K may significantly limit the flexibility of the routing mechanism—especially in long sequences where semantic dynamics can vary drastically.
> >
> > **A3.1:** Actually, the total budget $m \cdot K$ is the **same boundary** for both TopK and SeqTopK. The key difference is flexibility: Standard TopK is rigid, whereas SeqTopK treats this limit as a **cumulative budget**. This enables a **Banking Mechanism** that aligns perfectly with the semantic dynamics as you mentioned and we further elaborate in **A3.2** below.
> >
> > >This rigid upper bound could lead to under-utilization of expert capacity in later tokens, particularly when earlier tokens require fewer experts.
> >
> > **A3.2**:  On the contrary, SeqTopK tends to **allocate _more_ experts to later tokens and fewer to early tokens**, effectively creating a "**Banking Mechanism**". To verify this, we investigated the expert activation pattern versus sequence length in **Figure 7**.
> >
> > We observe a clear **upward trend** in the average number of activated experts as the sequence length grows. SeqTopK naturally assigns **early tokens (Savers)** with *fewer experts*, creating a **computational surplus.** Crucially, SeqTopK utilizes this surplus to assign **more experts** to **later tokens (Spenders)**, where context and semantic complexity are higher. We've add this analysis in **Appendix F**. Please let us know if there is more to clarify.
> >
> > (continue below)

---

> > > ### Author Response · Authors · 2025-11-21
> > >
> > > **Q4**: It remains unclear how well SeqTopK integrates with standard decoding techniques used in autoregressive language models, such as speculative decoding or beam search. Given the increasingly widespread use of such strategies for efficient inference, a discussion (or empirical validation) of SeqTopK’s compatibility with these paradigms would strengthen the practical relevance of the proposed method.
> > >
> > > **A4**:  SeqTopK is **fully compatible** with decoding techniques like speculative decoding and beam search as they are **structurally decoupled**. Decoding strategies operate on the **final output probability distribution (logits)**. In contrast, SeqTopK operates on **internal hidden states** within the MoE layers. So these concepts are **fully orthogonal** as they operate at different levels of the generation process. Follow you suggestion, we've included this discussion in **Section 5**.
> > >
> > > ---
> > > We hope that the explanations provided above adequately address your concerns. Following your valuable suggestions, we have revised and improved the manuscript accordingly. We would be grateful to know whether you find the updated version satisfactory, and we remain very willing to clarify any additional questions you may have.

---

### Official Review · Reviewer_nmgF · 2025-11-01

**Soundness:** 3
**Presentation:** 3
**Contribution:** 3
**Rating:** 6
**Confidence:** 2

**Summary:**

This paper proposes SeqTopK, a simple modification to standard TopK routing in MoE models that shifts expert budget allocation from the token level to the sequence level. Instead of assigning a fixed number of K experts to each token, SeqTopK selects the top T·K experts across all T tokens in a sequence, enabling context-aware dynamic allocation where harder tokens receive more experts and easier tokens receive fewer, while maintaining the same overall computational budget. The method requires minimal code changes, introduces no additional parameters or hyperparameters, and can be directly applied to pretrained MoE models through fine-tuning. Experimental results across math, coding, legal, and summarization tasks demonstrate consistent improvements over standard TopK routing, with particularly substantial gains (up to 16.9%) under higher sparsity regimes, suggesting SeqTopK is well-suited for next-generation ultra-sparse MoE architectures.

**Strengths:**

- The paper presents a creative and elegant solution by reframing the routing problem from token-level to sequence-level competition, allowing tokens to adaptively share expert budgets based on their relative difficulty, which appears to be a novel perspective in the MoE literature.

- The experimental evaluation demonstrates reasonable comprehensiveness across multiple base models (OLMoE, Qwen1.5), diverse tasks (math, coding, legal, summarization), and various sparsity levels, with consistent improvements that become more pronounced under higher sparsity conditions, suggesting robust empirical validation.

- The presentation is generally clear and well-structured, with effective visualizations (especially Figure 1's code comparison and Figure 2's token heterogeneity analysis) that help readers quickly grasp the core intuition and the minimal implementation changes required.

**Weaknesses:**

- The experimental evaluation excludes several recent adaptive MoE methods (e.g., MoE++, DynMoE mentioned but not compared), and the dismissal of these baselines based on architectural differences weakens the claim that SeqTopK represents the best parameter-free adaptive routing approach.

- The paper does not discuss scenarios where SeqTopK might underperform or fail, such as tasks requiring uniform token processing or sequences with atypical length distributions, limiting understanding of the method's applicability boundaries.

- The paper lacks systematic ablation experiments on critical design choices such as the impact of different token-level bound values, the necessity of lower/upper bounds, and how performance varies with different sequence length distributions during training versus inference.

- While Section 3.3 dismisses BatchTopK for language modeling, the experimental comparison in Tables 1-2 only reports "best performance" without showing the full sensitivity analysis or explaining under what conditions BatchTopK might still be competitive or preferable.

**Questions:**

- Could the authors provide theoretical analysis or formal guarantees explaining why sequence-level routing should outperform token-level routing? Additionally, can you identify and discuss specific scenarios or task characteristics where SeqTopK might underperform compared to standard TopK (e.g., sequences requiring uniform computation across all tokens, or tasks with highly variable sequence lengths)?

- he paper enforces per-token bounds (at least 1 expert, at most Ktok+2 experts) to prevent degenerate allocations. Could you provide systematic ablation studies showing: (a) performance with different bound values, (b) what happens without these bounds across different tasks, and (c) how sensitive SeqTopK is to sequence length variations during training versus inference?

- While the paper mentions MoE++ and DynMoE but excludes them from comparison due to architectural differences (additional zero/gate experts), could you clarify whether SeqTopK's gains would persist when compared against these methods? Additionally, for the BatchTopK comparison, could you provide full sensitivity analysis across different batch sizes rather than only reporting "best performance," to better understand when BatchTopK might be competitive?

---

> ### Author Response · Authors · 2025-11-21
>
> We sincerely appreciate your constructive comments and thoughtful suggestions, which are highly valuable for further improving the quality of our paper. We have carefully considered all the points you raised and have addressed each of your concerns in detail below.
>
> ---
> **Q1:** The experimental evaluation excludes several recent adaptive MoE methods (e.g., MoE++, DynMoE mentioned but not compared), and the dismissal of these baselines based on architectural differences weakens the claim that SeqTopK represents the best parameter-free adaptive routing approach.
>
> **A1:**  Follow your suggestion, we've added comparison between MoE++ in fine-tuning setting and found that SeqTopK **outperforms** MoE++ by a massive margin (**46.09** vs 18.91). Following [1], we implemented MoE++ on top of the pre-trained OLMoE model by adding one "zero" and one "copy" experts, and then fine-tuned the model under the _exact same_ hyperparameters used for SeqTopK.
>
> *Table 1:* Comparison with MoE++ (fine-tuned) on the GSM8K benchmark.
>
> | Methods | GSM8K     |
> | :-----: | --------- |
> |  base   | 15.58     |
> |  TopK   | 44.11     |
> |  MoE++  | 18.91     |
> | SeqTopK | **46.09** |
>
> As highlighted in _Table 1_, SeqTopK establishes a substantial **+143%** performance advantage over MoE++.  While MoE++ yields a marginal improvement over the base model (18.91 vs. 15.58), it is hindered by **fundamental architectural incompatibilities**, which means it is *only designed for pre-training from scratch.* In contrast, SeqTopK integrates seamlessly with existing pre-trained checkpoints, delivering large gains without re-training from scratch.
>
> Given the prohibitive computational cost of pre-training from scratch (approx. **10 days on 8x A100s**) and the strict rebuttal window, we restricted our scope to the fine-tuning setting. We defer the full pre-training comparison to future work and have detailed these fine-tuning results in **Appendix G**.
>
> [1] Jin P, Zhu B, Yuan L, et al. Moe++: Accelerating mixture-of-experts methods with zero-computation experts. arXiv preprint arXiv:2410.07348, 2024.
>
> ---
> **Q2**: The paper does not discuss scenarios where SeqTopK might underperform or fail, such as tasks requiring uniform token processing or sequences with atypical length distributions, limiting understanding of the method's applicability boundaries.
>
> **A2：** Currently, SeqTopK have two boundaries and limitations:
> - SeqTopK is specialized for *sparse MoE routing* and is inapplicable to dense architectures (which lack routing mechanisms). However, the core principle of _adaptive budget allocation_ is generalizable to other domains, such as dynamic attention pruning, feature selection, etc.
> - SeqTopK introduces *slight additional overhead*.  As shown in **Table 4**, throughput decreases by only $\approx $ 1\%(141.23 vs. 139.41 tokens/s), with marginal increases in memory and training time (<1%).  We note that this minor latency can be further mitigated via hardware-aware optimization (e.g., custom CUDA kernels), which we leave for future work.
>
>  We consider these limitations to be **negligible in practice**, as the slight overhead is significantly outweighed by the substantial performance gains (**+5.9%** in Table 1 and **+3.6%** in Table 2). The computational cost is virtually imperceptible (<1\%), and the architectural boundary represents a focused design choice rather than a fundamental flaw. Following your suggestion, we have incorporated this detailed discussion into **Appendix J**.
>
> (continue below)

---

> > ### Author Response · Authors · 2025-11-21
> >
> > **Q3**: The paper lacks systematic ablation experiments on critical design choices such as the impact of different token-level bound values, the necessity of lower/upper bounds, and how performance varies with different sequence length distributions during training versus inference.
> >
> > > The paper lacks systematic ablation experiments on critical design choices such as the impact of different token-level bound values,the necessity of lower/upper bounds.
> >
> > **A3.1**:  We sweep the upper bound from *K+1* to *no cap* and find that **$K+2$ provides the optimal trade-off** between flexibility and regularity across varying benchmarks. The results of ablation is presented in *Table 2* below:
> >
> > *Table 2 .* Ablations of upper bound selection of SeqTopK on OLMoE.
> >
> > | Methods | Lower Bound | Upper Bound | GSM8K     | MBPP      | HumanEval |
> > | :-----: | :------------: | :---------: | --------- | --------- | --------- |
> > |  TopK   |       K        |      K      | 44.11     | 21.04     | 13.41     |
> > | SeqTopK |       1        |     K+1     | 46.07     | 22.38     | 14.63     |
> > | SeqTopK |       1        |     K+2     | **46.09** | **23.21** | **15.24** |
> > | SeqTopK |       1        |     K+3     | 45.15     | 22.51     | **15.24** |
> > | SeqTopK |       1        |     K+4     | 44.54     | 22.00     | 15.01     |
> > | SeqTopK |       1        |      -      | 44.09     | 21.80     | 14.02     |
> > |||||||
> >
> > We find that while SeqTopK’s performance exhibits slight fluctuations(e.g., 14.02 to 15.24 on HumanEval) depending on the specific upper bound chosen, it consistently delivers **robust performance improvement** in most cases. Based on these empirical results, we recommend **$K+2$** as a robust default that yields significant gains without the need for task-specific tuning. We have included this ablation study and sensitivity analysis in **Appendix D** of the revised manuscript to provide transparency on hyperparameter selection.
> >
> > >How performance varies with different sequence length distributions during training versus inference
> >
> > **A3.2**:  As you suggested, we analyzed the model's robustness to varying sequence lengths during inference. We observe that SeqTopK **does not introduce additional fragility** to length variations.
> >
> > *Table 3 .* Performance vs. Inference Answer Length (OLMoE on GSM8K)
> >
> > | Answer length |  [0,50)   | [50,100)  | [100,150) | [150,250) | [250,) |
> > | :-----------: | :-------: | :-------: | :-------: | :-------: | :----: |
> > |   TopK Acc    |   60.48   |   51.78   |   33.71   |   25.08   |  0.0   |
> > |  SeqTopK Acc  | **63.72** | **52.99** | **35.41** |   25.04   |  0.0   |
> > |||||
> >
> > As *Table 3* shows, SeqTopK **consistently outperforms** the Top-K baseline across short, medium, and long contexts (e.g., **+2.8%** for lengths [100,250)).
> > While accuracy for both methods naturally declines as length increases (reflecting increased reasoning complexity), SeqTopK's degradation trend **closely tracks** the baseline, indicates that our dynamic routing mechanism is robust to length variations. The observed sensitivity is attributable to the intrinsic *limits of the pre-trained* model rather than the routing strategy.
> >
> > We note that evaluating sensitivity to _training_ length variations requires pre-training from scratch on varying context windows. Due to the limited rebuttal window and prohibitive cost (approx. **10 days on 8x A100s** for one run) , we focus our analysis on **inference robustness**, evaluating how the model behaves across different output lengths *within* its effective operating range. We've included this discussion in **Appendix H**.

---

> > > ### Author Response · Authors · 2025-11-21
> > >
> > > **Q4:** While Section 3.3 dismisses BatchTopK for language modeling, the experimental comparison in Tables 1-2 only reports "best performance" without showing the full sensitivity analysis or explaining under what conditions BatchTopK might still be competitive or preferable.
> > >
> > > **A4**: In fact, we have included the sensitivity analysis regrading to BatchTopK in Figure 4(b) of original submission (now **Fig 6(b)** in the revised version), demonstrating that  Batch-TopK is **highly sensitive to the evaluation batch size.** Batch-TopK achieves peak accuracy when the training and evaluation batch sizes match but degrades as the evaluation batch size increases (e.g., from 44.80 to 43.19). Therefore, we reported its best-case performance in the main table. Even in this optimal scenario for BatchTopK, SeqTopK consistently delivers **better performance**.
> > >
> > > (continue below)

---

> > > > ### Author Response · Authors · 2025-11-21
> > > >
> > > > **Q5**: Could the authors provide theoretical analysis or formal guarantees explaining why sequence-level routing should outperform token-level routing? Additionally, can you identify and discuss specific scenarios or task characteristics where SeqTopK might underperform compared to standard TopK (e.g., sequences requiring uniform computation across all tokens, or tasks with highly variable sequence lengths)?
> > > >
> > > > > Could the authors provide theoretical analysis or formal guarantees explaining why sequence-level routing should outperform token-level routing?
> > > >
> > > > **A5.1:**   Following your constructive suggestion, we have **theoretically** proven that under **token heterogeneity**, **SeqTopK consistently outperforms static TopK given the same expert budget.** We provide a detailed analysis from both empirical and theoretical perspectives:
> > > >
> > > > - **Empirical Motivation: Token Heterogeneity.**  As illustrated in **Figure 2(a)**, we observe that **not all tokens require the same expert budget.** When reducing the routing budget from $K=8$ to $K=4$, over *$60\%$* of tokens exhibit _negligible degradation_ in their predicted likelihood $P(x_t \mid x_{<t})$. In contrast, approximately $15\%$ of tokens suffer a severe drop of more than 0.5. This indicates that static routing (TopK) is **inefficient**: *it over-serves easy tokens while under-serving hard ones.* This token heterogeneity directly motivates our dynamic routing algorithm to reallocate expert resources.
> > > > - **Theoretical Insights: Optimality via Gradient Alignment.** Build on token heterogeneity, we prove that the **SeqTopK theoretically enjoys a tighter loss bound** compared to static TopK.
> > > > 	- **Formulation**: We model the routing decision as a constrained optimization problem (analogous to logistic regression), where the "resource"  is the number of active features $m_i$  allocated to each token $x_i$ to minimize its loss $\ell_i(m_i)$.
> > > > 	- **Optimality:** We derive that the global optimum requires the **marginal gain to be equalized** across all tokens:
> > > > 	  $$\frac{\partial \ell_i}{\partial m_i} + \lambda = 0 \implies \ell_i'(m_i^*) = -\lambda, \quad \forall i$$
> > > > 	- **Conclusion**: A uniform allocation strategy (i.e., static TopK) is inherently **suboptimal.** Since **token heterogeneity** implies that the optimal $m_i^*$ varies across tokens, fixing $m_i = K$ prevents the equality of marginal gradients. In contrast, SeqTopK adaptively varies $m_i$, allowing the model to satisfy this optimal gradient condition and thereby achieving superior resource utilization.
> > > >
> > > > In summary, SeqTopK succeeds not merely by heuristic design, but dynamically shifts computational resources from "easy" samples to "hard" samples, thereby satisfying the optimal gradient condition that static methods fundamentally violate. The detailed theoretical proof is provided in **Appendix E.**
> > > >
> > > > > Additionally, can you identify and discuss specific scenarios or task characteristics where SeqTopK might underperform compared to standard TopK (e.g., sequences requiring uniform computation across all tokens, or tasks with highly variable sequence lengths)?
> > > >
> > > > **A5.2:**  We've involved an additional analysis regarding to sequence length in **A3.2**. For other limitations and boundaries. Currently, SeqTopK have two boundaries and limitations:
> > > > 1. SeqTopK is specialized for *sparse MoE routing* and is inapplicable to dense architectures (which lack routing mechanisms). However, the core principle of _adaptive budget allocation_ is generalizable to other domains, such as dynamic attention pruning, feature selection, etc.
> > > > 2. SeqTopK introduces *slight additional overhead*.  As shown in **Table 4**, throughput decreases by only $\approx $ 1%(141.23 vs. 139.41 tokens/s), with marginal increases in memory and training time (<1%).  We note that this minor latency can be further mitigated via hardware-aware optimization (e.g., custom CUDA kernels), which we leave for future work.
> > > >
> > > >  We consider these limitations to be **negligible in practice**, as the slight overhead is significantly outweighed by the substantial performance gains (**+5.9%** in Table 1 and **+3.6%** in Table 2). The computational cost is virtually imperceptible (<1%), and the architectural boundary represents a focused design choice rather than a fundamental flaw. Following your suggestion, we have incorporated this detailed discussion into **Appendix J**.
> > > >
> > > > (continue below)

---

> > > > > ### Author Response · Authors · 2025-11-21
> > > > >
> > > > > **Q6**: The paper enforces per-token bounds (at least 1 expert, at most $K_{tok}$+2 experts) to prevent degenerate allocations. Could you provide systematic ablation studies showing: (a) performance with different bound values, (b) what happens without these bounds across different tasks, and (c) how sensitive SeqTopK is to sequence length variations during training versus inference？
> > > > >
> > > > > >Could you provide systematic ablation studies showing: (a) performance with different bound values, (b) what happens without these bounds across different tasks
> > > > >
> > > > > **A6.1:**  Please see **A3.1** .
> > > > >
> > > > > >how sensitive SeqTopK is to sequence length variations during training versus inference？
> > > > >
> > > > > **A6.2:** Please see **A3.2**.
> > > > >
> > > > > ---
> > > > > **Q7**:
> > > > > While the paper mentions MoE++ and DynMoE but excludes them from comparison due to architectural differences (additional zero/gate experts), could you clarify whether SeqTopK's gains would persist when compared against these methods? Additionally, for the BatchTopK comparison, could you provide full sensitivity analysis across different batch sizes rather than only reporting "best performance," to better understand when BatchTopK might be competitive
> > > > >
> > > > > >Could you clarify whether SeqTopK's gains would persist when compared against these methods?
> > > > >
> > > > > **A7.1**: Please see **A1**.
> > > > >
> > > > > >Additionally, for the BatchTopK comparison, could you provide full sensitivity analysis across different batch sizes rather than only reporting "best performance," to better understand when BatchTopK might be competitive?
> > > > >
> > > > > **A7.2:** Please see **A4.**
> > > > >
> > > > > ---
> > > > > Thank you very much for your insightful questions. We have carefully addressed all of your concerns above and have included additional experimental results in the revised manuscript, which we believe substantially strengthens and completes the work. We respectfully hope that these updates will allow you to re-evaluate our contribution in a more favorable light. Please do not hesitate to let us know if any further clarification would be helpful -- we would be more than happy to elaborate.

---

> ### Comment · Reviewer_nmgF · 2025-11-27
> **Thanks for your response**
>
> I thank the authors for their detailed response. I have read the new data and the responses. The ablation studies on upper bounds (Table 2 in the response) and the clarification regarding BatchTopK's sensitivity (Figure 6b) satisfactorily address my concerns regarding hyperparameter robustness and the baseline selection logic. The theoretical analysis in Appendix E also adds value to the submission.
>
> However, before I finalize my assessment, I have two follow-up inquiries to ensure the validity of the results and the method's robustness:
>
> - Regarding the MoE++ Fine-tuning Results:
> The reported performance drop for MoE++ (from 44.11 TopK to 18.91 MoE++) is extremely drastic. While I understand that MoE++ introduces new architectural components ("zero" and "copy" experts) that are typically trained from scratch, a drop to near-random performance suggests a potential initialization issue during fine-tuning. So my questions are:
>   - *How were the parameters for the newly added "zero" and "copy" experts initialized before fine-tuning? Were they randomly initialized, or was there a strategy to align them with the pre-trained latent space? If they were random, the comparison might be slightly unfair as the model has to "re-learn" routing while fighting noise, whereas SeqTopK inherits a stable state.*
>
> - Training-Inference Mismatch (Global vs. Causal Competition): I appreciate the inference length analysis (Table 3 in your rebuttal), but I am interested in the structural mismatch between training and inference. During training (offline), SeqTopK performs a *global* comparison where token $t_1$ competes with token $t_T$ for the budget. However, during autoregressive inference (online), token $t_1$ is generated when the cache is empty, and it cannot compete with future tokens.  My questions are:
>   -  *Does this mismatch lead to a distribution shift where early tokens in a generated sequence consistently consume more experts (since the budget $m \cdot K$ is not yet "stressed" by future hard tokens) compared to how they behaved during training? Did you observe any degradation in the quality of the *initial* tokens of generated sequences due to this lack of future context competition?*
>
> I look forward to your clarification on these points.

---

> > ### Author Response · Authors · 2025-11-27
> >
> > Thank you for your prompt response and clarifying your remaining concerns! We are happy to address them point by point below.
> >
> > ---
> > **Q1:** How were the parameters for the newly added "zero" and "copy" experts initialized before fine-tuning? Were they randomly initialized, or was there a strategy to align them with the pre-trained latent space? If they were random, the comparison might be slightly unfair as the model has to "re-learn" routing while fighting noise, whereas SeqTopK inherits a stable state.
> >
> > > How were the parameters for the newly added "zero" and "copy" experts initialized before fine-tuning? Were they randomly initialized, or was there a strategy to align them with the pre-trained latent space?
> >
> > **A1.1:** Good point! In fact, **there are no additional parameters for "zero" and "copy" expert.** Refering to MoE++ paper, these two experts can be  formulated as:
> >
> > - Zero Experts $E_{zero}$ : $E_{zero}(x)=0$. For any input $x$, it outputs a constant zero.
> > - Copy Experts $E_{copy}$ : $E_{copy}(x) = x$. For any input $x$, it outputs the identical $x$ as a "copy" operation.
> >
> > Therefore, the only newly added parameters reside in the **Gating Network** $W_{gate}$, which **expands from $\mathbb{R}^{N \times D}$ to $\mathbb{R}^{(N+2) \times D}$ to accommodate the two new routing choices.** Here, $N$ represents *the total number of experts* and $D$ is the hidden dimension of each token. We will discuss how to init $W_{gate}$ in **A1.2**.
> >
> >
> > >If they were random, the comparison might be slightly unfair as the model has to "re-learn" routing while fighting noise, whereas SeqTopK inherits a stable state.
> >
> > **A1.2:** Let $W_{org} \in \mathbb{R}^{N \times D}$ denote the original gate network parameters and $W_{new} \in \mathbb{R}^{(N+2) \times D}$ denote the new one. denote the new one. We implemented two strategies to initialize $W_{\text{new}}$:
> >
> > - **Random Initialization**: We randomly initialized the entire $W_{\text{new}}$ matrix. This approach caused immediate **model collapse**, as it completely discarded the pre-trained routing priors, leading to severe instability and divergence.
> > - **Partial Inheritance (Warm-Start)**: We initialized $W_{\text{new}}$ by explicitly **inheriting the pre-trained weights**. Specifically, the first $N$ rows were copied directly from $W_{\text{org}}$ ($W_{\text{new}}[:N, :] = W_{\text{org}}$), while the two newly added rows (for Zero and Copy experts) were initialized using the mean statistics of $W_{\text{org}}$. The results are shown in **Table 1** of the rebuttal.
> >
> > Even with this "Warm-Start" strategy designed to align with the pre-trained latent space, *the model still cannot recover from routing distribution shift within a fine-tuning setting* (e.g., training for 200 steps). In contrast, SeqTopK adapts seamlessly because it alters the selection scope **without modifying the router architecture**. This implementation simplicity enables context-aware routing in both fine-tuning (Tables 1 & 2) and pre-training (Table 3) scenarios. Furthermore, our method is orthogonal to architectural changes; in principle, **SeqTopK can even be applied with MoE++ (or other MoE variants) to further optimize their routing behaviors.**
> >
> > ---
> > **Q2:** However, during autoregressive inference (online), token $t_1$ is generated when the cache is empty, and it cannot compete with future tokens. My questions are:Does this mismatch lead to a distribution shift where early tokens in a generated sequence consistently consume more experts (since the budget is not yet "stressed" by future hard tokens) compared to how they behaved during training? Did you observe any degradation in the quality of the initial tokens of generated sequences due to this lack of future context competition?
> >
> > **A2:** This is an intresting question! Actually, when generating $t_1$, **the cache is not empty** cause the very first forward pass will encode pre-fix(eg, the problem description in GSM8K) into expert cache and $t_1$ is routed based on this.
> >
> > >Does this mismatch lead to a distribution shift where early tokens in a generated sequence consistently consume more experts (since the budget is not yet "stressed" by future hard tokens) compared to how they behaved during training?
> >
> > On the contrary, **SeqTopK tends to allocate more experts to later tokens and fewer to early tokens,** effectively creating a "Banking Mechanism". As shown in **Figure 7**, we investigated the expert activation pattern versus sequence length on GSM8K.
> >
> > We observe a **clear upward trend** in the average number of activated experts as the sequence length grows. SeqTopK naturally assigns **early tokens (Savers)** with fewer experts, creating a **computational surplus**. Crucially, SeqTopK utilizes this surplus to assign more experts to **later tokens (Spenders)**, where context and semantic complexity are higher.
> >
> > ---
> > We hope the responses provided adequately address your concerns. Please don’t hesitate to reach out if any further clarification is needed.

---

### Official Review · Reviewer_zvZz · 2025-11-01

**Soundness:** 3
**Presentation:** 3
**Contribution:** 2
**Rating:** 4
**Confidence:** 4

**Summary:**

This paper introduces SeqTopK, a context-level routing strategy for Mixture-of-Experts (MoE) models that allocates experts across sequences rather than individual tokens. The authors argue that traditional token-level TopK routing is suboptimal due to its fixed expert budget per token, and propose SeqTopK to enable dynamic, context-aware expert allocation while maintaining the same overall computational budget. The method includes both offline and online variants, the latter designed for autoregressive decoding via an "Expert Cache."

**Strengths:**

The idea of shifting expert allocation from token-level to sequence-level is both intuitive and innovative. SeqTopK is implemented with minimal code changes, introduces no new parameters, and maintains full compatibility with pre-trained MoE models, making it practical. The paper provides comprehensive experiments across multiple domains and model architectures, consistently showing performance gains over TopK and prior adaptive methods.

**Weaknesses:**

While the paper presents a simple yet effective idea and demonstrates convincing performance improvements in its experiments, the core contribution—context-aware routing—lacks sufficient explanation and validation of its underlying mechanisms and implications. The paper excels at explaining the "what" (the method design and results) but falls short in deeply analyzing the "why" (the intrinsic workings) and the "under what conditions" (the method's boundaries and limitations).

**Questions:**

1.The paper mentions enforcing per-token bounds to prevent "degenerate allocations." This suggests that the original, unconstrained SeqTopK might suffer from training instability or severe load imbalance. These constraints are inherently heuristic. Could the authors clarify:
a)How were these specific bounds determined? Do they introduce hyperparameters that require tuning?

b) Compared to standard TopK, does SeqTopK (even with constraints) exhibit greater expert load variance or higher auxiliary balancing loss during training? Is this a hidden cost of its dynamic allocation capability?

2.A core idea of MoE is expert specialization. Fixed-budget Token-Level TopK provides each expert with relatively stable "exposure" from all tokens. In contrast, SeqTopK's dynamic allocation might lead to "easy" tokens consistently activating only a fixed, small subset of experts, while "hard" tokens competitively activate more experts. Could this exacerbate a "Matthew effect" among experts, causing some experts to become functionally degenerate due to consistently servicing easy patterns, while others suffer in generalization due to over-specialization on hard patterns? Have the authors analyzed the distribution of expert activation under SeqTopK and whether this impacts the degree of expert specialization?

3.The paper compares SeqTopK well against other MoE routing methods like MRL-TopK and BatchTopK. However, the idea of adaptive computation has been extensively explored in broader areas like Early Exiting and Token Pruning. SeqTopK essentially performs a horizontal computation allocation within the MoE layer. Have the authors considered conceptual or experimental comparisons with these vertical or cross-module adaptive methods (e.g., allowing difficult tokens to pass through more model layers)? Discussing SeqTopK's unique position and limitations within the "adaptive computation spectrum" would help provide a more comprehensive assessment of its contribution.

---

> ### Author Response · Authors · 2025-11-21
>
> We sincerely thank you for your thorough reading and thoughtful comments. We truly appreciate the time and care you invested in reviewing our work. Motivated by your insightful suggestions, we have substantially expanded the discussions and conducted additional experiments on SeqTopK. We deeply value your feedback, and we respectfully address each of your concerns point-by-point below. We hope our clarifications and revisions successfully resolve all the issues you raised and help strengthen the manuscript.
>
> ---
> **Q1:**  The paper excels at explaining the "what" (the method design and results) but falls short in deeply analyzing the "why" (the intrinsic workings).
>
> **A1:**  Following your constructive suggestion, we have **theoretically** proven that under **token heterogeneity**, **SeqTopK consistently outperforms static TopK given the same expert budget.** We provide a detailed analysis from both empirical and theoretical perspectives:
>
> - **Empirical Motivation: Token Heterogeneity.**  As illustrated in **Figure 2(a)**, we observe that **not all tokens require the same expert budget.** When reducing the routing budget from $K=8$ to $K=4$, over *$60\%$* of tokens exhibit _negligible degradation_ in their predicted likelihood $P(x_t \mid x_{<t})$. In contrast, approximately $15\%$ of tokens suffer a severe drop of more than 0.5. This indicates that static routing (TopK) is **inefficient**: *it over-serves easy tokens while under-serving hard ones.* This token heterogeneity directly motivates our dynamic routing algorithm to reallocate expert resources.
> - **Theoretical Insights: Optimality via Gradient Alignment.** Build on token heterogeneity, we prove that the **SeqTopK theoretically enjoys a tighter loss bound** compared to static TopK.
> 	- **Formulation**: We model the routing decision as a constrained optimization problem (analogous to logistic regression), where the "resource"  is the number of active features $m_i$  allocated to each token $x_i,$ to minimize its loss $\ell_i(m_i)$.
> 	- **Optimality:** We derive that the global optimum requires the **marginal gain to be equalized** across all tokens:
> 	  $$\frac{\partial \ell_i}{\partial m_i} + \lambda = 0 \implies \ell_i'(m_i^*) = -\lambda, \quad \forall i$$
> 	- **Conclusion**: A uniform allocation strategy (i.e., static TopK) is inherently **suboptimal.** Since **token heterogeneity** implies that the optimal $m_i^*$ varies across tokens, fixing $m_i = K$ prevents the equality of marginal gradients. In contrast, SeqTopK adaptively varies $m_i$, allowing the model to satisfy this optimal gradient condition and thereby achieving superior resource utilization.
>
> In summary, SeqTopK succeeds not merely by heuristic design, but dynamically shifts computational resources from "easy" samples to "hard" samples, thereby satisfying the optimal gradient condition that static methods fundamentally violate. The detailed theoretical proof is provided in **Appendix E.**
>
> ---
> **Q2:**  Lack of discussion on the method's boundaries and limitations
>
> **A2:**  Currently, SeqTopK have two boundaries and limitations:
> - SeqTopK is specialized for *sparse MoE routing* and is inapplicable to dense architectures (which lack routing mechanisms). However, the core principle of _adaptive budget allocation_ is generalizable to other domains, such as dynamic attention pruning, feature selection, etc.
> - SeqTopK introduces *slight additional overhead*.  As shown in **Table 4**, throughput decreases by only $\approx$ 1\% (141.23 vs. 139.41 tokens/s), with marginal increases in memory and training time (<1%).  We note that this minor latency can be further mitigated via hardware-aware optimization (e.g., custom CUDA kernels), which we leave for future work.
>
>  We consider these limitations to be **negligible in practice**, as the slight overhead is significantly outweighed by the substantial performance gains (**+5.9%** in Table 1 and **+3.6%** in Table 2). The computational cost is virtually imperceptible (<1%), and the architectural boundary represents a focused design choice rather than a fundamental flaw. Following your suggestion, we have incorporated this detailed discussion into **Appendix J**.
>
> (continue below)

---

> > ### Author Response · Authors · 2025-11-21
> >
> > **Q3:** The paper mentions enforcing per-token bounds to prevent "degenerate allocations." This suggests that the original, unconstrained SeqTopK might suffer from training instability or severe load imbalance.
> >
> > **A3:** In fact, the unconstrained SeqTopK **does not** suffer from training instability or severe load imbalance. To address this concern, we have added a comprehensive stability analysis as fine-tuning OLMoE on GSM8K dataset.
> >
> > As shown in **Figure 5** of the revised paper, the unconstrained SeqTopK maintains robust expert load variance that is comparable to the TopK baseline (**0.00027** vs 0.00032). Consequently, the upper bound serves not to mitigate instability, but to *prevent the model from becoming "over-reactive"* on certain downstream tasks and results in **lower expert variance (0.00021)**. Empirically, we find that imposing an upper bound consistently yields better performance than the unconstrained setting, as detailed in *Table 3* below.
> >
> > These findings conclusively demonstrate that our method is **robust by design**. This ensures that SeqTopK can be **seamlessly deployed** into existing MoE frameworks without the risk of training volatility. We have added a comprehensive stability analysis and discussion in **Figure 5** and **Section 4.4**.
> >
> > ---
> > **Q4**: These constraints are inherently heuristic. Could the authors clarify: a)How were these specific bounds determined? Do they introduce hyperparameters that require tuning?
> >
> > **A4**: The upper bond is determined **empirically**. We sweep the upper bound from *K+1* to *no cap* and find that **$K+2$ provides the optimal trade-off** between flexibility and regularity across varying benchmarks. The results of ablation is presented in *Table 3* below:
> >
> > *Table 3 .* Ablations of upper bound selection of SeqTopK on OLMoE.
> >
> > | Methods | Lower Bound | Upper Bound | GSM8K     | MBPP      | HumanEval |
> > | :-----: | :------------: | :---------: | --------- | --------- | --------- |
> > |  TopK   |       K        |      K      | 44.11     | 21.04     | 13.41     |
> > | SeqTopK |       1        |     K+1     | 46.07     | 22.38     | 14.63     |
> > | SeqTopK |       1        |     K+2     | **46.09** | **23.21** | **15.24** |
> > | SeqTopK |       1        |     K+3     | 45.15     | 22.51     | **15.24** |
> > | SeqTopK |       1        |     K+4     | 44.54     | 22.00     | 15.01     |
> > | SeqTopK |       1        |      -      | 44.09     | 21.80     | 14.02     |
> >
> > We find that while SeqTopK’s performance exhibits slight fluctuations(e.g., 14.02 to 15.24 on HumanEval) depending on the specific upper bound chosen, it consistently delivers **robust performance improvement** in most cases. Based on these empirical results, we recommend **$K+2$** as a robust default that yields significant gains without the need for task-specific tuning. We have included this ablation study and sensitivity analysis in **Appendix D** of the revised manuscript to provide transparency on hyperparameter selection.
> >
> > ---
> > **Q5:** Compared to standard TopK, does SeqTopK (even with constraints) exhibit greater expert load variance or higher auxiliary balancing loss during training? Is this a hidden cost of its dynamic allocation capability?
> >
> >
> > **A5:** No. In contrast, SeqTopK achieves **better load balancing.** As detailed in **A3** (and **Figure 5**), there is no "hidden cost." Even without constraints, SeqTopK maintains expert load variance competitive with the static Top-K baseline (e.g., **0.00027** vs **0.00032**). Furthermore, the upper bound acts as a regularizer, resulting in **lower variance and better expert utilization** than the baseline. We believe this improved balancing dynamics accounts for the observed performance gains. We have added this analysis in **Section 4.4.**
> >
> > ((continue below))

---

> > > ### Author Response · Authors · 2025-11-21
> > >
> > > **Q6:**  Could this exacerbate a "Matthew effect" among experts, causing some experts to become functionally degenerate due to consistently servicing easy patterns, while others suffer in generalization due to over-specialization on hard patterns? Have the authors analyzed the distribution of expert activation under SeqTopK and whether this impacts the degree of expert specialization?
> > >
> > > > Could this exacerbate a "Matthew effect" among experts?
> > >
> > > **A6.1:** No. On the contrary, empirical evidence demonstrates that **SeqTopK mitigates expert imbalance rather than exacerbating it**. To verify this, we present both training and evaluation metric as follows:
> > > - **Training Dynamics (Low Variance):** As detailed in **A3**, the expert load variance confirms that no collapse occurs. Even in the unconstrained setting, the variance is comparable to the static baseline. Furthermore, enforcing the **$K+2$ upper bound** actively prevents any single expert from dominating, resulting in _lower_ variance than standard TopK.
> > > - **Evaluation Metrics (High Entropy):** As shown in **Figure 4(a)**, SeqTopK consistently yields **higher Expert-Routing Entropy** compared to the baseline. This higher entropy quantitatively proves a broader, more uniform utilization of the expert pool.
> > >
> > > Metrics from both training and inference confirm that SeqTopK enforces a balanced distribution rather than a concentrated one. We attribute the observed performance gains to this improved equilibrium. We have added this discussion in **Section 4.4**.
> > >
> > >
> > > >Have the authors analyzed the distribution of expert activation under SeqTopK and whether this impacts the degree of expert specialization?
> > >
> > > **A6.2:** Good suggestion! Actually, SeqTopK promotes a significantly **more uniform utilization** of the expert pool. Our **Expert-load histogram in Figure 4(b)** provides clear visual evidence, showing that the standard TopK baseline suffers from a *"spiky"* distribution, whereas SeqTopK enables a visibly *smoother* and more even expert load distribution.
> > >
> > > By allowing "hard" tokens to access _more_ experts and "easy" tokens _fewer_. SeqTopK **relieves the congestion on experts**, ensuring a more diverse and equitable exposure across the expert pool. We have added this discussion in **Section 4.4**.
> > >
> > > (continue below)

---

> > > > ### Author Response · Authors · 2025-11-21
> > > >
> > > > **Q7:**  Have the authors considered conceptual or experimental comparisons with these vertical or cross-module adaptive methods (e.g., allowing difficult tokens to pass through more model layers)? Discussing SeqTopK's unique position and limitations within the "adaptive computation spectrum" would help provide a more comprehensive assessment of its contribution.
> > > >
> > > >  >Discussing SeqTopK's unique position within the "adaptive computation spectrum" would help provide a more comprehensive assessment of its contribution.
> > > >
> > > > **A7.1:** Indeed, SeqTopK shares a holistic motivation with broader adaptive methods—addressing **token heterogeneity**. To clarify our contribution, we categorize these adaptive approaches into a **"Three-Dimensional Spectrum"** and position SeqTopK as the pioneer of the "Horizontal" dimension:
> > > > - **Depth Adaptivity (Vertical):** Methods like _Early Exiting_ [1] or _Mixture-of-Depths_ [4,5] dynamically allocate computation across network depth (e.g., skipping layers).
> > > > - **Length Adaptivity (Sequence):** Methods like _Token Pruning_ (e.g., ToMe [2], SparseVLM [3]) reduce sequence length by dropping or merging tokens to save compute.
> > > > - **SeqTopK (Horizontal/Width):** SeqTopK fills a critical gap by introducing **Horizontal Adaptivity**. Instead of skipping layers (Depth) or dropping tokens (Length), it operates **within** the MoE layer, dynamically adjusting the _number of active parameters_ (experts) per token.
> > > >
> > > > By occupying the "Horizontal" niche, SeqTopK offers distinct benefits:
> > > > - **Parameter-Free Integration:** Unlike vertical/sequence methods that often require architectural changes, SeqTopK is parameter-free and can be easily plugged into existing MoE frameworks.
> > > > - **Orthogonality:** Since SeqTopK optimizes _width_, it is mathematically orthogonal to _depth_ and _length_ methods. It can be seamlessly stacked with them to achieve a fully adaptive **"3D efficient architecture."** For instance, one could use token pruning to reduce sequence length, and then apply SeqTopK to optimally route the remaining tokens.
> > > >
> > > > We have expanded **Section 5** to include this comparative framework and discussion.
> > > >
> > > > > Discussing SeqTopK's limitations.
> > > >
> > > > **A7.2:**  As presented in **A2**, SeqTopK is currently specialized for MoE architectures and introduces a marginal overhead ($\approx 1\%$ throughput drop). However, we consider these trade-offs to be **negligible in practice**, as the slight cost is significantly outweighed by the substantial performance gains (**+5.9%** in Table 1 and **+3.6%** in Table 2). Following your suggestion, we have incorporated this detailed discussion into **Appendix J**.
> > > >
> > > > [1] Teerapittayanon, Surat, Bradley McDanel, and Hsiang-Tsung Kung. "Branchynet: Fast inference via early exiting from deep neural networks." _2016 23rd international conference on pattern recognition (ICPR)_. IEEE, 2016.
> > > >
> > > > [2] Bolya, Daniel, et al. "Token Merging: Your ViT But Faster." _The Eleventh International Conference on Learning Representations_.
> > > >
> > > > [3] Zhang, Yuan, et al. "SparseVLM: Visual Token Sparsification for Efficient Vision-Language Model Inference." Forty-second International Conference on Machine Learning. SparseVLM
> > > >
> > > > [4] Raposo, David, et al. "Mixture-of-depths: Dynamically allocating compute in transformer-based language models." _arXiv preprint arXiv:2404.02258_ (2024).
> > > >
> > > > [5] Bae, Sangmin, et al. "Mixture-of-recursions: Learning dynamic recursive depths for adaptive token-level computation." _arXiv preprint arXiv:2507.10524_ (2025).
> > > >
> > > > ---
> > > > We are sincerely grateful for your thoughtful and constructive feedback. Your comments have been invaluable in guiding us to further improve and refine the manuscript. We truly appreciate the time and expertise you devoted to reviewing our work, and we would be more than happy to provide any additional clarification should further questions arise.We are genuinely grateful for your thoughtful feedback, which has been instrumental in helping us refine the manuscript. Please do not hesitate to reach out if you have any additional comments or questions.

---

### Public Comment · ~Giang_Do1 · 2025-11-21
**Note on Overlap with Earlier arXiv Work**

Thank you for this submission. For transparency, I would like to point out that our earlier work, public on arXiv before the submission deadline, covers a very similar SMoE Router Mechanism with this submission.

- Unified Sparse Mixture of Experts, arXiv:2503.22996, posted on Mar 29, 2025 (https://huggingface.co/papers/2503.22996)

Readers may find the overlap in motivation, methodology, and results relevant when evaluating novelty and contribution. Thanks.

---

> ### Author Response · Authors · 2025-11-21
>
> We thank the authors of Unified Sparse Mixture of Experts (USMoE) for bringing this work to our attention. While both works aim to improve upon the TopK routing paradigm, we must clarify that **they address fundamentally different problems through distinct mechanisms**:
>
> - **Distinct Motivation: Token Heterogeneity (Ours) vs. Routing Stability**
>     - **SeqTopK (Ours)**: Our primary motivation is **Token Heterogeneity**—the premise that **not all tokens require same expert budget**. We address this by dynamically allocating expert capacity, **assigning more experts to "hard" tokens and fewer to "easy" tokens within a sequence** to optimize efficiency.
>     - **USMoE:** On the other hand, USMoE is motivated by **Routing Stability** and **preventing representation collapse**. Their goal is to find a mathematical "middle ground" between Token Choice (which suffers from load imbalance) and Expert Choice (which suffers from information leakage).
> - **Distinct Mechanism: Selection Scope (Ours) vs. Scoring Function**
>     - **SeqTopK (Scope Shift):** keeps the *standard scoring function* but **changes the selection scope**, moving from a local token-level constraint to **a global sequence-level budget.**
>     - **USMoE (Scoring Modification):** In contrast, USMoE **fundamentally modifies the routing formulation** itself. They propose a "Unified Score" that combines *Token Choice* and *Expert Choice* scores ($(1-\alpha) \cdot f_{token} + \alpha \cdot f_{expert}$) while maintaining **a token-level selection mechanism**. This represents an architectural modification to the router's internal logic, whereas SeqTopK focuses on the dynamic reallocation of the token-expert budget.
>
> We believe this clarifies that the **two methods are distinct in both motivation and method.**
> We have integrated this comparative discussion into the **Related Work** section of the revised manuscript. Please let us know if there are any further details we can clarify regarding this comparison.

---

> > ### Public Comment · ~Giang_Do1 · 2025-11-21
> >
> > Dear Authors,
> >
> > Thank you for the constructive and thoughtful discussion. To help readers clearly understand the relationship between USMoE and this submission, we would like to summarize the comparison across three aspects:
> > 1. **Token/Expert Selection Mechanism**:
> > Both works employ a Top $T.K$ selection over tokens dimension and experts dimension, and the practical implementation is highly similar.
> > This can be observed by comparing USMoE Algorithm 1 with Figure 1 in the submission.
> > 2. **Router Scoring Function**:
> > USMoE introduces a unified routing score that interpolates between Token Choice and Expert Choice.
> > The scoring function in the submission corresponds to a special case of the USMoE unified score when $\alpha$ = 0.
> > 3. **Motivation - Dynamic Expert Allocation**:
> > Both USMoE and the submission aim to address dynamic expert allocation.
> > In USMoE, we show that some "noisy" tokens require fewer experts,
> > which is equivalent to allocating more experts to other tokens under a fixed compute budget.
> >
> > We would like to sincerely note that readers may find the overlap in **Token/Expert Selection Mechanism** and **Routing formulation** relevant when evaluating the novelty of the submission.
> > Thank you again for your engagement and clarification.

---

### Meta-Review · Area_Chair_kb5g · 2026-01-06

**Summary:**

The authors proposes SeqTopK, a routing strategy for moe models that shifts the expert budget from a fixed per-token constraint to a sequence-level budget. This allows for dynamic expert allocation based on token difficulty. While the reviewers acknowledged the intuition of handling token heterogeneity and the strong empirical results in fine-tuning settings, there are significant concerns regarding the soundness and theoretical grounding of the proposed method. Specifically, the discrepancy between the training objective, which uses global sequence context, and the autoregressive inference mechanism remains a critical issue. Additionally, the reliance on heuristic bounds to prevent expert collapse and the limited evaluation scope (fine-tuning vs. pre-training from scratch) weaken the claims of the paper.

**Reviewer Concerns:**

There are three primary remaining concerns raised by reviewers:
1. Online Inference Mechanism, @Reviewer atcY and nmgF: This is the most critical outstanding issue. Reviewer atcY pointed out a potential flaw where the online algorithm, lacking access to future tokens, cannot truly perform "sequence-level" optimization. While the authors argued for a "banking mechanism" (saving budget on early tokens), this introduces a distribution shift between training (global competition) and inference (local dynamic thresholding). The theoretical guarantee that this does not degenerate to a heuristic variant of TopK is weak.
2. Heuristic Nature & Stability @ Reviewer zvZz: The method relies on enforcing hard constraints to prevent degenerate allocations. Reviewer zvZz noted that this suggests the core routing objective is not inherently stable or self-regulating, and the "why" behind the specific bounds remains empirical rather than theoretical.
3. Evaluation Scope: The validation is primarily on fine-tuning pre-trained models. As noted by reviewers, routing algorithms often behave differently when trained from scratch (where representation collapse is a higher risk). The lack of full pre-training experiments leaves the long-term stability of SeqTopK unproven.

**Reviewer Scores:**

I think reviewers with low scores will slightly raise their scores because some issues have been addressed; however, the overall score should still be around the average, and due to the high competition within the batch, it will still be below the acceptance line.

---

### Decision · Program_Chairs · 2026-01-26

Reject